# Overcoming size limits with dynamic templates enabling large area single crystal nanowire arrays for photodetectors

Mingjie Feng[1,2] ✉, Jiwon Byun [3], Zongbao Li[4], Zhiqiang Xie[1], Wenbo Lu[5], Xin Wen[5], Liang Ding[5], Tingting Wu[5], Sumbal Jamshaid[6], Klaus Götz [7], Chaohui Li [1,2], Zijian Peng[1,2], Huiying Hu[1,2], Jingjing Tian [1,2], Jack Elia [1], Tobias Unruh [7], Marcus Halik [3], Ding-Jiang Xue [5] ✉, Andres Osvet[1] ✉ & Christoph J. Brabec [1,8] ✉

Ordered one dimensional perovskite single-crystal nanowire arrays, which combine high surface-to-volume ratios, directional charge transport, and mechanical flexibility, are typically prepared through solution or vapor phase techniques using templates of silicon, polydimethylsiloxane, photoresist, or aluminum oxide to control crystal growth. However, the size limits of these templates restrict the scalability of the arrays. Here, we introduce a dynamic template-assisted coating strategy that integrates blade coating to address this limitation. The method enables deposition of nanowire arrays on substrates with an area 12 times larger than the template. Incorporating a fluorinated passivating agent into the precursor suppresses surface defect formation and improves structural quality. Photodetectors based on MAPbBr$_3$ arrays achieve a detectivity of $3.9 \times 10^{14}$ Jones, a linear dynamic range of 160.3 dB, and a responsivity of 1660 A W$^{-1}$, and retain 90.3% of their photocurrent after 300 h at 85% relative humidity without encapsulation.

Compared to traditional silicon-based photodetectors, organic-inorganic hybrid perovskites (OIHPs), particularly methylammonium lead trihalides (MAPbX$_3$, where MA = methylammonium and X = I, Br, or Cl), are considered even more ideal materials for fabricating photodetectors due to their lower manufacturing cost[1], superior light absorption properties[2], tunable electronic band gaps[3,4], and enhanced compatibility with flexible integration[5,6]. Single-crystal perovskites, free from grain boundaries, exhibit higher carrier mobility, longer diffusion lengths, and extended lifetimes due to significantly reduced

trap state densities, making them a focal point of research[7,8]. Currently, a wide variety of single-crystal structures are available for the fabrication of photodetectors, ranging from zero-dimensional (0D) to three-dimensional (3D) forms[9–18]. Among them, photodetectors based on 1D single-crystal structures demonstrate superior performance and are more extensively studied. This can be attributed to the highly anisotropic properties and increased surface area of 1D micro/nanowires, which provide efficient pathways for directional carrier transport[19], while simultaneously reducing carrier recombination[12],

[1]Department of Materials Science and Engineering, Institute of Materials for Electronics and Energy Technology (i-MEET), Friedrich-Alexander-Universität Erlangen-Nürnberg, Erlangen, Germany. [2]Erlangen Graduate School in Advanced Optical Technologies (SAOT), Erlangen, Germany. [3]Department of Materials Science, Organic Materials and Devices, Interdisciplinary Center for Nanostructured Films (IZNF), Friedrich-Alexander-Universität Erlangen-Nürnberg, Erlangen, Germany. [4]School of Materials Science and Engineering, Wuhan Textile University, Wuhan, China. [5]Beijing National Laboratory for Molecular Sciences (BNLMS), CAS Key Laboratory of Molecular Nanostructure and Nanotechnology, Institute of Chemistry, Chinese Academy of Sciences, Beijing, China. [6]Department of Materials Science and Engineering, Friedrich-Alexander-Universität Erlangen-Nürnberg, Erlangen, Germany. [7]Institute for Crystallography and Structural Physics, Friedrich-Alexander-Universität Erlangen-Nürnberg, Erlangen, Germany. [8]Helmholtz-Institute Erlangen-Nürnberg for Renewable Energy (HI ERN), Erlangen, Germany. ✉e-mail: mingjie.feng@fau.de; djxue@iccas.ac.cn; andres.osvet@fau.de; christoph.brabec@fau.de

thereby enhancing photoresponse speed and detection sensitivity. Moreover, due to their high aspect ratio, these structures exhibit remarkable mechanical flexibility, showing great potential for applications in the fabrication of flexible devices[5,20].

To date, two primary methods for fabricating 1D single-crystal nanowire arrays (SCNWAs) have been reported: template-free methods (such as solution-phase techniques, vapor deposition, and hot-injection methods) and template-assisted methods[21–23]. Comparatively, template-assisted methods utilize pre-designed channel patterns to precisely control the nucleation and growth of perovskite nanowires, allowing them to be arranged into ordered arrays at specified locations. Consequently, it has become the preferred choice for researchers involved in 1D SCNWAs fabrication and investigation. Looking ahead, the assembly of large-area, high-quality perovskite micro/nanowire arrays is critical for advancing integrated devices, including active matrix displays, logic circuits, and image sensors[20,24]. However, scaling up perovskite nanostructures via template-assisted methods remains a significant challenge, as the fabrication of larger templates introduces heightened complexity and cost[25,26].

We turn our attention to the blade-coating method, a widely adopted large-area thin-film fabrication technique known for its simple equipment structure, high material utilization efficiency, and strong compatibility with roll-to-roll production lines, making it extensively used in industrial printing[27,28]. Unfortunately, this approach is not sufficiently space-constrained, as the turbulent transport of perovskite solutes during coating leads to their random accumulation in the supersaturated phase, resulting in uneven nucleation and crystallization[29,30]. Jie et al. successfully fabricated 1 cm-long MAPbI$_3$ microwire arrays with a certain degree of orientation by optimizing the coating process to balance nucleation and crystal growth within the precursor solution under blade guidance[20]. Subsequently, the same research group, along with Zhang and Kim et al., employed a microchannel-confined crystallization strategy to further enhance crystal alignment along the growth direction[31–33]. However, their approaches remain dependent on template size. Moreover, defects in perovskite single crystals are predominantly located at the surface, and the high surface-to-volume ratio of SCNWAs inevitably introduces a higher density of trap states[31,32]. Therefore, our primary focus is to address both the template size dependency and the issue of surface defects in the material.

Here, we propose a dynamic template-assisted coating (DTA) strategy to fabricate high-quality 1D-SCNWAs. During the coating process, a specially treated polydimethylsiloxane (PDMS) template moves on the substrate surface. The dynamic confined space generated by the template surface microchannels and the substrate facilitates the orderly growth of crystal self-assembly, and the capillary force ensures a stable supply of solutes throughout the process. To reduce defect density and enhance device stability, we in situ introduce a fluorine-containing passivating agent (methylamine trifluoroacetate, MTFA). After treatment, the defect density of the target device is reduced by more than an order of magnitude. Moreover, fluorine atoms contribute to a robust moisture barrier by forming a hydrophobic layer, enabling the device to retain 90.3% of its initial performance even after 300 h of exposure to 85% RH without encapsulation. The results show that photodetectors based on large-area MAPbBr$_3$ SCMWAs prepared via the DTA method exhibit high overall performance, achieving a detectivity of up to $3.9 \times 10^{14}$ Jones, a linear dynamic range of 160.3 dB, and a responsivity of 1660 A W$^{-1}$.

## Results and discussion
### Fabrication and characterization of perovskite nanowires
Figure 1a illustrates the fabrication of large-area perovskite SCNWAs using our DTA technique with a modified PDMS template. By adjusting the crosslinker ratio, we modify the PDMS to enhance its elastic modulus by nearly fivefold, ensuring stable crystal growth, maintaining substrate adhesion, and reducing friction during the process (Supplementary Fig. 1). To further minimize friction and prevent mold deformation, high-molecular-weight silicone oil is incorporated (Supplementary Fig. 2), while an increased solvent contact angle (~102°) facilitates smooth demolding (Supplementary Fig. 3). The PDMS mold, featuring periodic wide and shallow channels defined by an SU-8 photolithography template (Supplementary Fig. 4), was secured under the blade. The coating system operates under optimized parameters to achieve uniform and large-area SCNWAs, thereby eliminating dependence on template size and enabling the fabrication of SCNWAs with an area 12 times that of the template itself, which exceeds previously reported methods (as shown in Supplementary Table 1). Detailed experimental procedures and settings are provided in the Experimental Section and Supplementary Fig. 5.

We systematically investigate key factors influencing the morphology of perovskite SCNWAs, including PDMS template modifications, precursor solution concentration, template movement speed, and substrate temperature (Supplementary Figs. 6–7). Precise control of the evaporation rate is critical to maintaining the precursor concentration within the optimal range—between saturation and supersaturation—thereby ensuring crystallization without premature nucleation. Excessively high substrate temperatures or extreme precursor concentrations disrupt this balance, leading to abnormal nucleation and discontinuous arrays. Conversely, low temperatures or excessive speeds shift the precursor into the Landau-Levich regime, causing merging rather than crystallization upon template release[30]. Notably, this strategy demonstrates broad applicability, enabling fabrication on substrates with varying flexibility (e.g., silicon wafers and polyimide (PI) films, Fig. 1b) and extending to other MAPbX$_3$-based perovskite materials (Supplementary Fig. 8). We further explore the potential of the DTA strategy for lateral dimensional scaling by sequentially shifting the substrate and repeating the coating process. This approach enables the fabrication of multiple adjacent SCNWAs regions on a 4-inch wafer, as demonstrated in Supplementary Fig. 9.

We analyze the crystallographic quality and morphology of MAPbBr$_3$ SCNWAs fabricated via the DTA strategy. The SCNWAs exhibit an absorption cutoff at ~555 nm and a PL emission peak at ~540 nm (Supplementary Fig. 10), corresponding to an optical bandgap of 2.23 eV, which is consistent with single-crystal values[34]. Fluorescence imaging under 405 nm excitation reveals uniformly distributed green lines, indicating well-aligned SCNWAs without cracks, while black regions confirm precise growth confinement by the PDMS template (Fig. 1c). SEM images (Fig. 1d) reveal a uniform, defect-free morphology conforming to the microchannel dimensions. EDS mapping (Fig. 1e) demonstrates a homogeneous distribution of Pb and Br, with an atomic ratio of ~1:3 (Supplementary Fig. 11). TEM analysis (Fig. 1f) further confirms a dense structure with sharp edges, while SAED patterns (inset) display sharp, discrete spots, verifying the single-crystal nature. The (100) pole figure displays a distinct central diffraction spot without ring-like or line-like clusters, confirming a highly consistent (001) orientation. This aligns with the XRD pattern, where peaks at $2\theta = 15.2°$ and 30.3° correspond to the (001) and (002) planes, respectively, contrasting with non-template-coated samples (Supplementary Fig. 12). Additionally, the (101) pole figure exhibits fourfold symmetric diffraction spots, reflecting the single-crystal structure's symmetry (Fig. 1g)[35,36]. Collectively, these results validate the successful fabrication of high-quality perovskite SCNWAs via the DTA process.

The large surface area of SCNWAs makes them particularly susceptible to defect-induced carrier recombination and material degradation[37]. To address this issue, we investigate surface defects in MAPbBr$_3$ single crystals, focusing on bromine vacancies (V$_{Br}$), which readily form at surfaces or grain boundaries due to their low formation energy[38,39]. These vacancies leave uncoordinated Pb$^{2+}$ ions that act as non-radiative recombination centers, while their mobility further

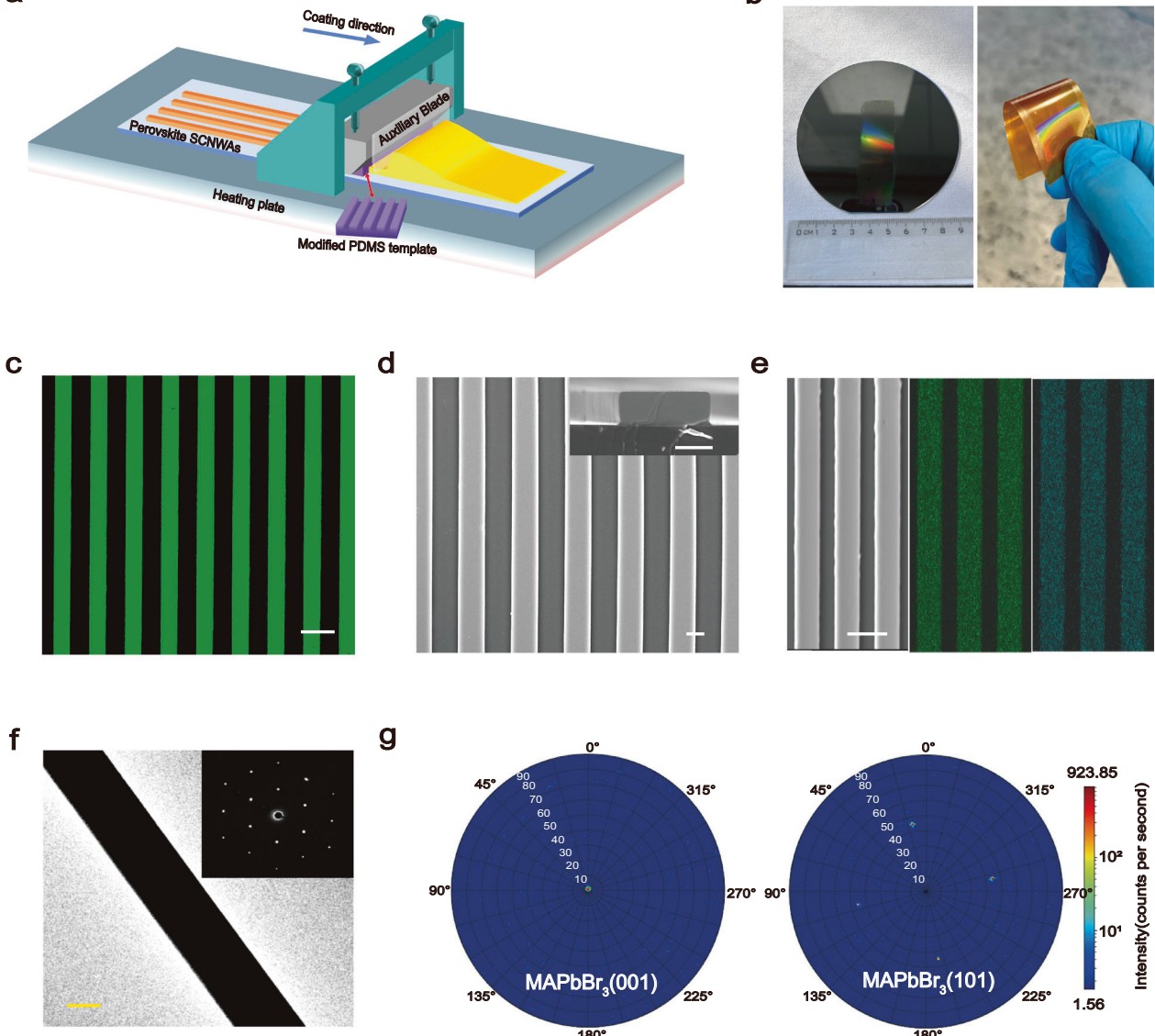

**Fig. 1 | MAPbX₃ SCNWAs preparation mechanism demonstration and related characterization. a** Schematic diagram of the self-assembly of MAPbX₃ SCNWAs on the substrate surface by the dynamic confinement coating strategy. **b** Photographs of large-area MAPbBr₃ SCNWAs on a Si wafer and PI film. **c** Fluorescence micrograph of large-area MAPbBr₃ SCNWAs. Scale bar = 20 μm. **d** Top-view and cross-sectional Scanning electron microscope (SEM) images of MAPbBr₃ SCNWAs. Scale bar = 1 μm. **e** The corresponding energy dispersive spectrometer (EDS) elemental mapping of the MAPbBr₃ SCNWAs. Scale bar = 10 μm. **f** Typical transmission electron microscopy (TEM) image of a single MAPbBr₃ SCNWAs and corresponding selected-area electron diffraction (SAED) pattern. Scale bar = 1 μm. **g** X-ray diffraction (XRD) pole figure measured along the (001) and (101) directions of the MAPbBr₃ SCNWAs.

degrades the material's optoelectronic performance and stability. More critically, the loss of more than two Br⁻ ions from $PbBr_6$ octahedra can lead to the formation of Pb-Pb dimers, introducing deep-level defects that exacerbate degradation[40]. Recent studies have shown that chemical passivation through strong bonding interactions can stabilize surface Pb sites and suppress Pb dimer formation[41,42], offering a promising strategy to mitigate these challenges.

### Effects of methylamine trifluoroacetate on perovskite nanowires

To mitigate the typical surface defects in MAPbBr₃ SCNWAs—including undercoordinated $Pb^{2+}$ ions, Pb–Pb dimers, and $MA^+$ vacancies—we seek a passivation agent capable of simultaneously interacting with both metal-related and A-site-related defect species. MTFA is selected due to its unique dual-functionality: its TFA⁻ anion contains a Lewis-basic carboxylate group and highly electronegative –CF₃ groups, which can engage in complementary interactions with different types of defects. Even though this type of passivating agent—as well as related TFA⁻-based molecules—has been successfully employed to improve crystallinity and stability in polycrystalline perovskite films[43,44], its application to single-crystalline materials, particularly 1D-SCNWAs, remains largely unexplored, yet appears to involve an intriguing and potentially significant underlying mechanism. To bridge this gap, we incorporate MTFA into the precursor solution. Density functional theory (DFT) calculations reveal that the carboxylate group (O=C–O⁻) of TFA⁻ acts as a strong electron-donating Lewis base due to its high electronegativity (Fig. 2a). As illustrated in Fig. 2b, TFA⁻ coordinates with undercoordinated $Pb^{2+}$ ions, stabilizing Pb sites and suppressing Pb dimer formation. Additionally, fluorine atoms in the –CF₃ group form hydrogen bonds with $MA^+$ cations, stabilizing MA sites and reducing $V_{MA}$ density (Supplementary Fig. 13). Owing to its dual binding capability, TFA⁻ exhibits stronger adsorption on MABr-terminated

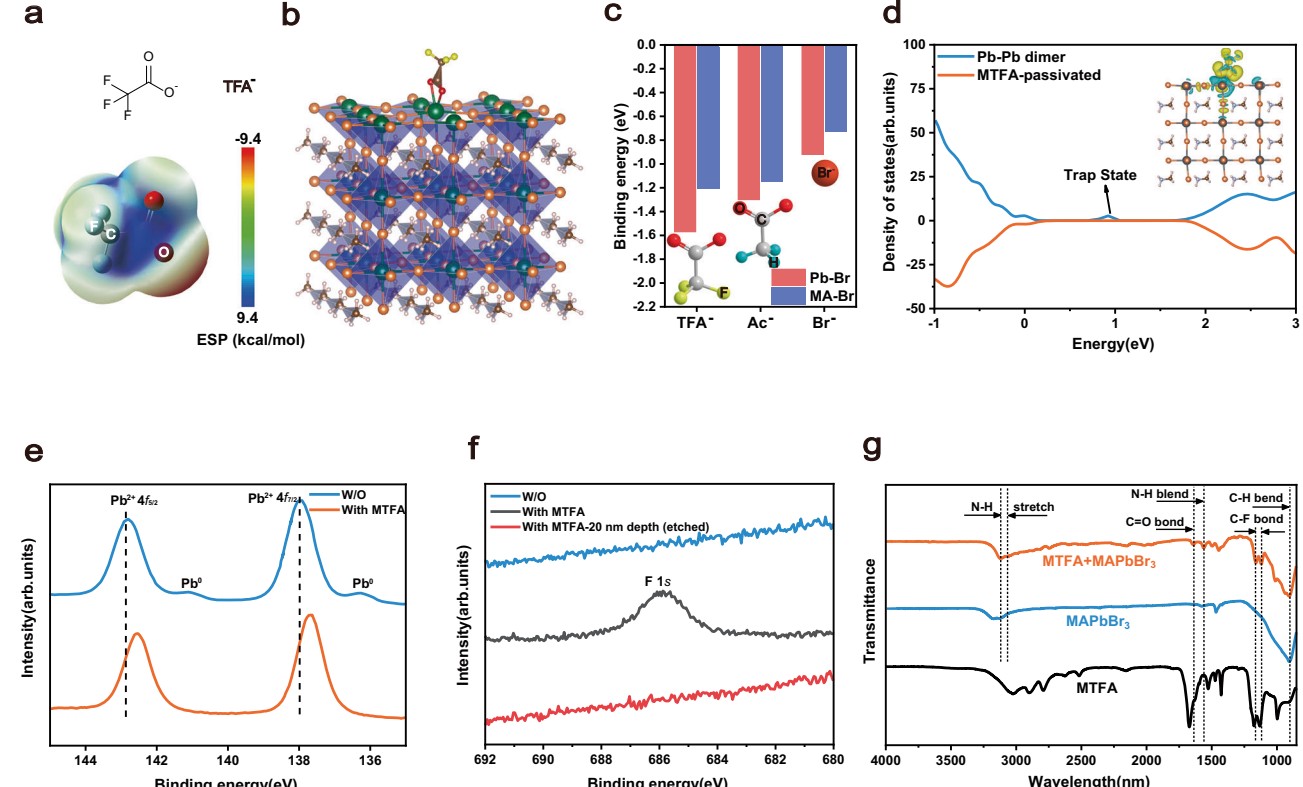

**Fig. 2 | Mechanism and Characterization of MTFA Passivation on MAPbBr₃ Single Crystal Nanowire Arrays. a** Electrostatic potential (ESP) distribution of the Lewis-base group of MTFA. **b** Diagram of the interaction between TFA⁻ and perovskite on t-PB surface. **c** The binding energy of Br⁻, Ac⁻ (acetate), and TFA⁻ at two different termination surfaces. **d** DOS of the perovskite surface with and without passivation. The X-ray photoelectron spectroscopy (XPS) characterization of MAPbBr₃ SCNWAs with and without MTFA passivation. **e** Pb 4f and (**f**) F ls. **g** The fourier-transform infrared spectroscopy (FTIR) spectra of MTFA alone, MAPbBr₃ with and without MTFA passivation.

surfaces (t-MB) than on PbBr₂-terminated surfaces (t-PB) (Fig. 2c). Binding energy calculations confirm that both Ac⁻ and TFA⁻ preferentially adhere to the crystal surface, with TFA⁻ showing superior affinity on t-MB. To validate these findings, we compare the photoluminescence quantum yield (PLQY) of devices treated with different anions (TFA⁻, Ac⁻, and Br⁻) while keeping the MA⁺ cation constant. Furthermore, density of states (DOS) calculations for Pb–Pb dimer defects on a t-PB surface reveal a deep trap state at 0.93 eV within the bandgap, indicating severe defects in unpassivated samples. After TFA⁻ passivation, this trap state is significantly suppressed (Fig. 2d), demonstrating its efficacy in mitigating defect-induced carrier trapping.

XPS is employed to investigate the interaction between TFA⁻ and the perovskite. The Pb 4f binding energy exhibits a negative after treatment (Fig. 2e), indicating electron donation from TFA⁻ oxygen atoms to Pb²⁺ 6p orbitals, which stabilizes the perovskite structure. In contrast, the control sample displays prominent Pb⁰ peaks at 136.6 eV and 141.5 eV, indicative of metallic lead defects caused by excessive $V_{Br}$. These defects act as charge traps, degrading device performance. MTFA treatment significantly reduced Pb⁰ content, likely due to Pb-O coordination suppressing defect formation. A distinct F 1s peak in MTFA-treated MAPbBr₃ SCNWAs disappears after etching, confirming that TFA⁻ primarily resides on the surface (Fig. 2f). FTIR further corroborates these interactions. The C = O stretching peak redshifts from 1671 to 1640 cm⁻¹, confirming TFA⁻ coordination with Pb²⁺. Additionally, the N–H bending (1573 cm⁻¹) and stretching (3177 cm⁻¹) peaks shift to lower wavenumbers, indicating hydrogen bonding between TFA⁻ fluorine atoms and organic cations. These interactions promote uniform TFA⁻ distribution, enhancing film homogeneity and defect passivation (Fig. 2g, Supplementary Fig. 14).

## Effect of passivation on optical and electrical properties

We first assess the crystal quality by measuring the X-ray rocking curve of the (001) plane (Fig. 3a). The absence of split peaks indicates no twin formation, and a full width at half maximum (FWHM) as low as 0.024° confirms significantly improved crystallinity. To investigate the passivation effect of MTFA on surface defects in SCNWAs, we further carry out PL and time-resolved PL (TRPL) measurements. MTFA-treated SCNWAs exhibit a markedly enhanced PL intensity compare to untreated samples, along with a blue shift in the PL peak from 543.6 to 540.8 nm, suggesting a substantial reduction in defect density (Fig. 3b). As shown in Supplementary Fig. 15, TFA⁻ treatment yields the highest PLQY, consistent with theoretical predictions. The TRPL decay curves are fitted using the perovskite carrier recombination simulator (PEARS) based on a bimolecular trapping-detrapping model, extracting rate constants for bimolecular recombination ($K_B$), capture ($K_T$), and decapture ($K_D$), as well as trap density ($N_t$) (Fig. 3c, Supplementary Table 2)[45,46]. The results reveal that MTFA effectively passivates surface defects, reducing defect density by nearly an order of magnitude and doubling the PLQY to a maximum of 9.9%. In addition, we calculate the carrier lifetimes of the samples before and after MTFA treatment (Supplementary Table 3). The MAPbBr₃ microwire arrays treated with MTFA exhibit significantly extended average carrier lifetimes at low excitation energy densities ($\tau_{avg}$ = 134.6 ns, compared to 91.2 ns for the untreated sample at 58.9 nJ cm⁻², Supplementary Table 3), indicating that non-radiative recombination is suppressed. Interestingly, we find that $\tau_{avg}$ of the unpassivated sample first increased and then decreased with increasing energy density, which was due to trap filling followed by bimolecular or Auger recombination, while that of the passivated sample showed a steady downward trend, which was consistent with

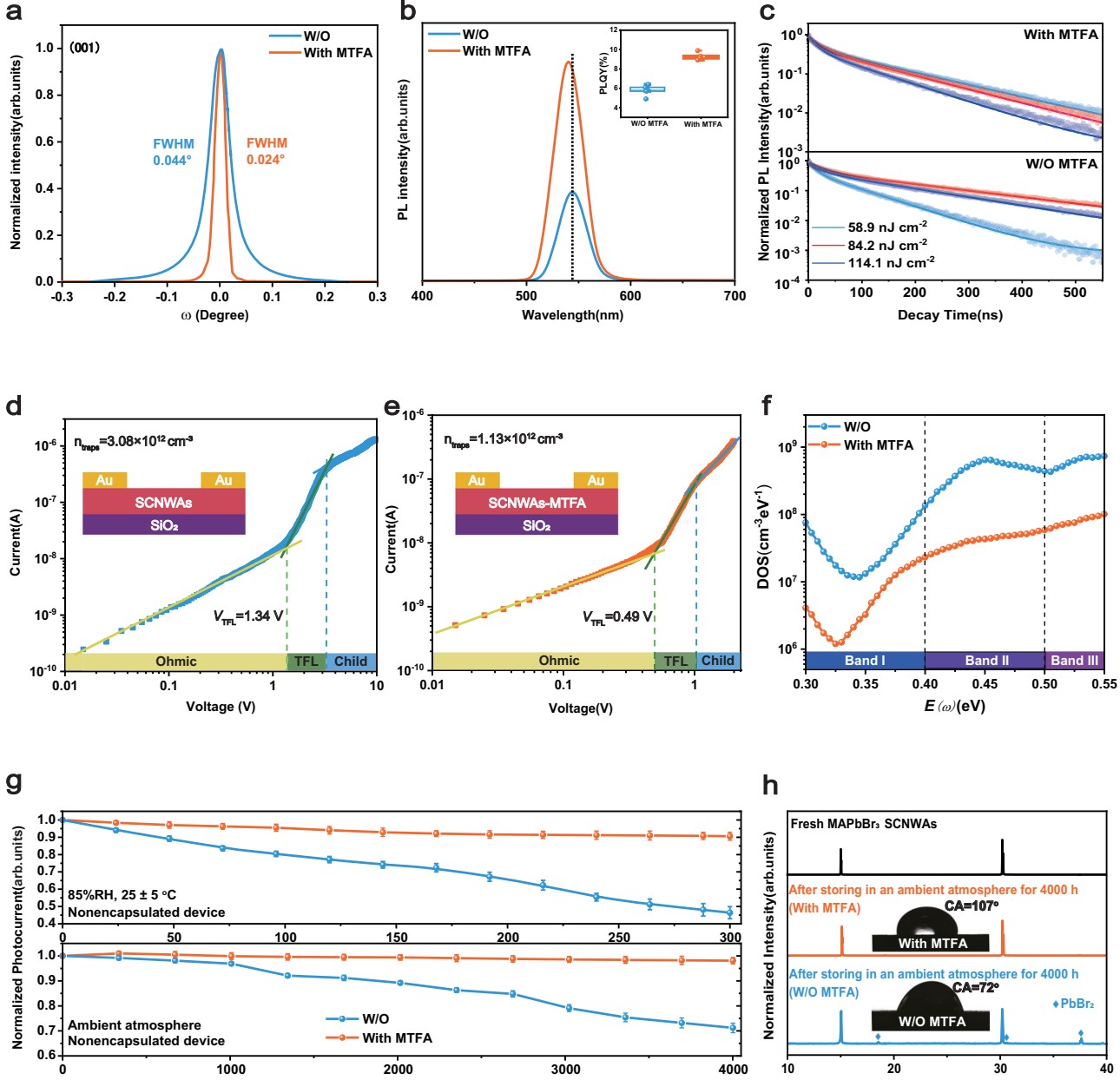

**Fig. 3 | Characterization of optical and electrical properties before and after passivation. a** High-resolution XRD rocking curves of the (001) crystal planes of MAPbBr$_3$ SCNWAs with and without MTFA treatment. **b** PL, PLQY, and **c** TRPL spectra of MAPbBr$_3$ SCNWAs with and without MTFA treatment. Space charge limited current (SCLC) measurement on (**d**) control and **e** MTFA-treated MAPbBr$_3$ SCNWAs. **f** Trap-density-of-states (tDOS) curves of MAPbBr$_3$ SCNWAs with and without MTFA treatment. **g** Comparison of the changes in the photocurrent (normalized) of MTFA-treated and untreated photodetectors when exposed to indoor air for 4000 h and to RH = 85% for 300 h. Error bars represent standard deviations from three independent measurements. **h** XRD patterns of the MAPbBr$_3$ SCNWAs with MTFA treatment before and after being exposed to air for 4000 h. The contact angles (CA) of water on control and MTFA-treated samples are shown in the inset.

the suppression of trap saturation. These behaviors are consistent with previous reports on high-quality perovskite films[47]. To further quantify defect reduction, we perform SCLC measurements using a hole-only lateral device (Au/SCNWAs/Au). Dark $J-V$ characteristics (Fig. 3d, e) showed negligible hysteresis at 10,000 mV s$^{-1}$, as confirmed by scan rate-dependent measurements (Supplementary Fig. 16). We attribute this behavior to the mixed ionic–electronic conduction nature of perovskites, where ion migration under an electric field can distort SCLC measurements. At low scan rates, mobile ions have sufficient time to redistribute, resulting in significant hysteresis. By contrast, increasing the scan rate effectively suppresses ion movement and trap-state relaxation. When the scan rate reaches 10,000 mV s$^{-1}$, the

hysteresis effect becomes nearly negligible, as mobile ions no longer have sufficient time to respond to the external electric field[48,49]. The $N_t$ is calculated using the formula $V_{TFL} = qN_tL^2/2\varepsilon_0\varepsilon_r$, where $\varepsilon_r$ is the relative dielectric constant (25.5)[50], $\varepsilon_0$ is the vacuum permittivity (8.85×10$^{-14}$ F cm$^{-1}$), $q$ is the elementary charge (1.60 × 10$^{-19}$ C), $V_{TFL}$ is the trap-filled limit voltage, and $L$ is the electrode spacing (3.5 × 10$^{-3}$ cm). The calculated $N_t$ is 1.13 × 10$^{12}$ cm$^{-3}$, which is comparable to that of many bulk MAPbBr$_3$ single crystals, indicating the optimization of SCNWAs quality by MTFA treatment[51].

Thermal admittance spectroscopy (TAS) reveals a significant reduction in $N_t$ across all defect levels in MTFA-treated SCNWAs[40](Fig. 3f). Band I (0.30–0.40 eV), associated with shallow

traps, shows a reduction to ~10% of the control level, consistent with suppressed bromide loss and aligning with DFT predictions. More notably, Band II (0.40–0.45 eV) and Band III (0.45–0.50 eV), corresponding to deeper-level defects, exhibit an approximately one-order-of-magnitude decrease in trap density (tDOS), highlighting MTFA's effectiveness in mitigating these more detrimental trap states. Beyond defect passivation, MTFA also influenced SCNWAs' morphology. Untreated SCNWAs become discontinuous when the microchannel width exceeds 7 μm, whereas MTFA-treated samples retained uniform and continuous arrays even at widths up to 10 μm (Supplementary Fig. 17). This improvement is attributed to enhanced ion diffusion within narrow microchannels, ensuring a stable solute supply for crystal growth, as well as surface tension-regulated solvent flow along the substrate, promoting uninterrupted crystallization[14]. The optimal MTFA-to-MAPbBr$_3$ mass ratio is determined to be 1:20 (Supplementary Fig. 18). Demonstrating the spatial uniformity of crystal quality and optoelectronic properties is essential for validating the scalability and practical applicability of large-area SCNWAs. Spatially resolved PL and transient response measurements are therefore performed at 43 locations across an area of approximately 20 mm × 60 mm (Supplementary Fig. 19a). The 3D pseudocolor PL map (Supplementary Fig. 19b) shows minimal peak shift (standard deviation: 1.4%), while the corresponding intensity plot (Supplementary Fig. 19c) reveals consistent spectral shapes (FWHM variation: 2.4%, Supplementary Table 4). Supplementary Movie 1 further confirms the structural consistency over a 50 mm serial scan. Additionally, transient photoresponse data exhibit uniformly stable rise and decay behaviors across the sampled positions, highlighting the performance and spatial uniformity of the SCNWAs.

Despite the excellent optoelectronic properties of perovskite-based devices, their long-term application is hindered by instability under water and oxygen exposure. To evaluate MTFA's role in enhancing stability, we test the long-term air and short-term high-humidity stability of unencapsulated MAPbBr$_3$ SCNWAs. As shown in Fig. 3g, the MTFA-treated device retained 98% of the initial photocurrent after more than 4000 h of air exposure, and no obvious PbBr$_2$ peak is observed in the XRD spectrum (Fig. 3h). Even under 85% RH for 300 h, the devices maintain 90% of their initial photocurrent. We attribute this enhanced stability to the highly ordered microstructure of MAPbBr$_3$ SCNWAs, which inherently improves hydrophobicity[6], combined with the hydrophobic fluorine-rich groups of TFA⁻ forming a uniform protective layer on the surface, effectively blocking moisture infiltration. This is further supported by the contact angle measurements, with the inset of Fig. 3h showing a 35° increase in hydrophobicity for the treated sample.

## Photodetector performance and application in nanowire arrays

We demonstrate the excellent crystal quality of large-area MAPbBr$_3$ SCNWAs achieved via the DTA strategy. To evaluate their photodetection potential, we fabricate a lateral-configuration photodetector, as illustrated in the inset of Fig. 3e. The current–voltage ($I$–$V$) characteristics under dark conditions and 405 nm illumination (0–100 μW cm⁻²) reveal that the dark current is about dozens of pA at 4 V bias, attributed to effective defect suppression (Fig. 4a). With increasing light intensity, the photoresponse becomes pronounced, and the photocurrent is significantly enhanced. The light intensity-dependent current density and responsivity (Fig. 4b) demonstrate the device's excellent capability to discriminate between varying illumination levels. Notably, the photodetector achieves a maximum responsivity of 1660 A W⁻¹ at 5 V bias, representing high performance among 1D SCNWAs-based photodetectors (Supplementary Table 5). Responsivity ($R$) is defined as $R = J_{ph}/L_{light}$, where $J_{ph}$ is the photocurrent density and $L_{light}$ is the incident light intensity. A wide linear dynamic range (LDR) is critical for image sensors, ensuring stable operation across a broad range of light intensities without

signal distortion. LDR is expressed as LDR = 20 log ($I_{upper}/I_{lower}$), where $I_{upper}$ and $I_{lower}$ represent the maximum and minimum photocurrents within the linear range, respectively. Using the noise current as the minimum detectable current, the device achieves an LDR of 160.3 dB at 5 V bias—the highest reported value for 1D SCNWAs-based photodetectors to date (Supplementary Table 5). This performance surpasses silicon photodetectors (120 dB) and significantly exceeds other widely used photodetectors, such as InGaAs (66 dB)[52].

The $D^*$, a key metric for evaluating a photodetector's ability to detect weak light signals, is calculated as $D^* = R(Af)^{1/2}/I_{noise}$, where $A$ is the device area, $f$ is the bandwidth, and $I_{noise}$ is the noise current (Fig. 4c). Owing to the suppressed noise level and enhanced photocurrent generation, our device achieves a remarkable $D^*$ of 3.9 ×10¹⁴ Jones at 5 V bias voltage. Supplementary Figs. 20 and 21 show the measured $R$, LDR, $D^*$, and noise current of the device under different bias voltages (1 V and 10 V), together with the instrumental noise baseline. While a higher bias significantly improves $R$ (2424 A W⁻¹), it also increases the noise current, resulting in a sharp decline in $D^*$ to only 6.8% of its value at 5 V. Response speed, characterized by rise time ($\tau_{rise}$, 10% to 90% of maximum photocurrent) and decay time ($\tau_{decay}$, 90% to 10%), is another critical parameter. As shown in Fig. 4d, under an irradiance of 50 mW cm⁻² and 5 V, the samples exhibit a rapid response with $\tau_{rise}$ of 98.7 μs and $\tau_{decay}$ of 56.6 μs, which is significantly faster than those reported in recent studies on similar types of devices (Supplementary Table 5).

Stability is a critical performance metric for perovskite photodetectors. To evaluate this, we measure the photocurrent response under extreme humidity conditions (85% RH) at room temperature. Continuous monitoring over 300 h reveals a minimal photocurrent degradation of only 7% after 150 h and ~13% after 300 h (Fig. 4e). This excellent stability is attributed to the hydrophobic effect induced by the MTFA passivating agent. To further assess device performance, we investigated photoresponse uniformity and imaging capabilities. A large-area detection array (20 × 60 mm²) is fabricated on a MAPbBr$_3$ SCNWAs absorbing layer, comprising 10 × 50 detectors as pixels. Dark current measurements, conducted by randomly selecting five columns along the X-axis (Fig. 4f), revealed remarkably low coefficients of variation, indicating excellent uniformity. Slightly higher fluctuations in the first and last columns are likely due to edge effects during the coating process. For imaging evaluation, a metal mask with an IMEET pattern is placed between a uniform illumination source and the photodetector array. A 525 nm light source projects the optical pattern onto the 500-pixel sensor array (Fig. 4g). The photocurrent distribution, compiled into a two-dimensional contrast map (Fig. 4h), accurately reproduces the optical pattern. Notably, MTFA-treated sensors exhibit significantly lower current fluctuations and more stable signals compared to the control group. With a response time of 0.099 ms—substantially faster than the human eye's recognition threshold (~42 ms)—this 1D SCNWA-based image sensor demonstrates considerable potential for advanced photodetection applications[53].

In summary, we introduce a blade-coating-based template-assisted strategy that eliminates dependence on template size, enabling the fabrication of large-area, ordered MAPbX$_3$ SCNWAs on various substrates. Using MAPbBr$_3$ as the research object, we apply DFT to identify deep-level defects and introduce the fluorinated passivator MTFA to the precursor solution. This passivator effectively reduces defect density and enhances stability. Photodetectors based on large-area MAPbBr$_3$ SCNWAs achieve high comprehensive performance (with $D^*$ of 3.9 ×10¹⁴ Jones, LDR of 160.3 dB, and $R$ of 1,660 A W⁻¹) and demonstrate excellent stability, maintaining 90.3% of their performance after 300 h at 85% RH without encapsulation. This method provides an alternative route for achieving high-performance perovskite SCNWAs-based photodetectors.

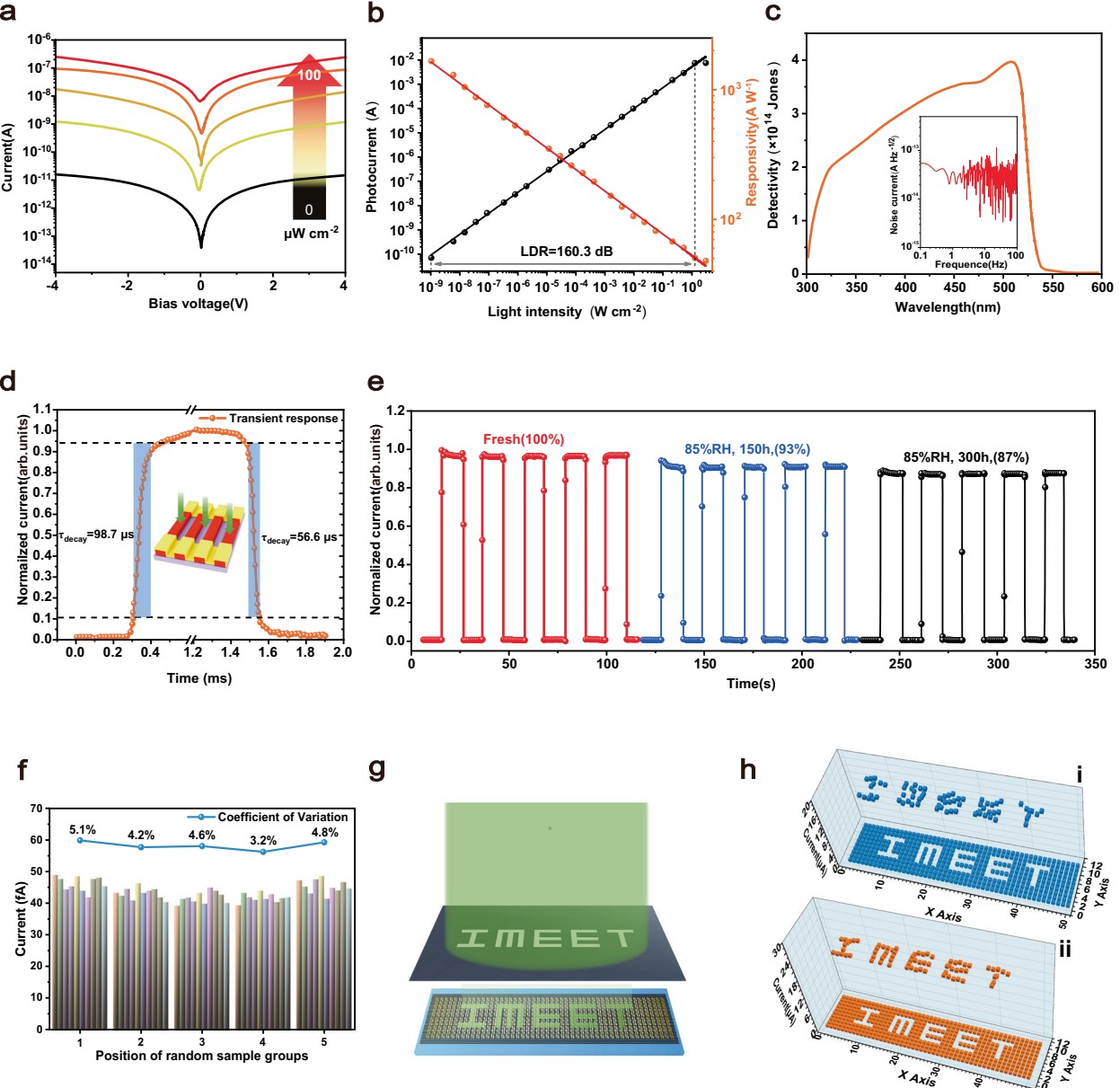

**Fig. 4 | Performance analysis of the photodetector based on the MAPbBr₃ single crystal nanowire arrays. a** $I$–$V$ curves of MAPbBr₃ SCNWAs in the dark state and in light illumination with different irradiant power (0, 0.1, 1, 10, 100 μW cm⁻²). **b** The illumination power-dependent photocurrents and responsivities. **c** Plots of specific detectivity of the device as a function of light wavelength at a fixed light intensity of 5 μW cm⁻² and frequency-dependent noise currents of microwire devices in the dark at 5 V. **d** The response speed of the device was 98.7 μs for rise and 56.6 μs for decay, and a schematic illustration of the photodetector. **e** Response of the device to the pulsed light of the same intensity after 0, 150, and 300 h in an environment with 85% RH. **f** The statistical results of dark current measurements along five randomly selected lines (columns one, fifteen, twenty-five, thirty-five, and fifty) in the 10 × 50 photodetector array and the coefficient of variation for each group. **g** Schematic illustration for the projection imaging mechanism. The light projects an optical pattern with contrast between light and dark areas onto the surface of the sensor unit through the mask. **h** The output current of each pixel measured under a 2 V bias with the I-MEET pattern (Fig. **i**, sensors without MTFA treatment; Fig. **ii**, sensors treated with MTFA).

## Methods

### Materials and solution preparation

The precursors for MAPbX₃ (X = Cl, Br, and I), including MAPbI₃, MAPbBr₃, and MAPbCl₃, were prepared by mixing PbX₂ (Tokyo Chemical Industry, >98.0%) solution with MAX (Sigma-Aldrich, 98%) in a mixed solvent of DMF or DMSO (Sigma-Aldrich, super dry reagent) at a concentration of 0.5 M. The PDMS precursor and cross-linking agent were purchased from Dow Corning (Sylgard 184). High-temperature, corrosion-resistant adhesive was obtained from Henkel (Germany) and used for fixing the template and auxiliary blade. Methylamine Trifluoroacetate (MTFA, >98.0%) and Methylamine Acetate (MAAc,

>98.0%) were purchased from TCI. For the passivation solution, MTFA, MAAc, or MABr were directly dissolved in 1 mL of MAPbBr₃ precursor solution, with a molar ratio of passivation agent to MAPbBr₃ of 5:21. The resulting solution was stirred at 50 °C for 6 h in an argon-filled glove box and then filtered for further use.

### Template preparation

Initially, SU-8 (MicroChem-2002) surface microchannel structures of varying widths were fabricated on a silicon substrate via photolithography to serve as templates for PDMS mold formation. Following this, the surface was treated with octadecyltrichlorosilane (OTS) to

impart hydrophobic properties. The PDMS precursor was prepared by mixing the base with a crosslinker at a mass ratio of 5:1, along with 2 wt % of high molecular weight silicone oil (cSt). This mixture was then cast onto the SU-8-patterned silicon substrate, degassed, and cured at 80 °C for 3 h in an oven. After curing, the PDMS mold was carefully demolded from the master template. The elevated crosslinker content and curing temperature enhance the modulus of the PDMS mold, thereby safeguarding the microchannels and reducing the friction coefficient against smooth substrates. The incorporation of high molecular weight silicone oil further decreases the friction coefficient while maintaining leak-free performance. A PDMS template with an area of 1 cm² (2 × 0.5 cm) was then fixed under the blade to implement the DTA strategy. The mechanical properties, specifically the elastic modulus of the PDMS, were evaluated using a universal testing machine (Zwick Z050). The tribological properties were examined using a ball-on-disk tribometer (Supplementary Fig. 2), employing glass balls with a diameter of 4 mm.

## Growth of single crystal nanowire arrays for photodetectors

Perovskite SCNWAs were fabricated on a smooth, rigid substrate using PDMS molds with varying widths of 2 μm, 5 μm, and 10 μm. Before the experiment, the blade coating apparatus was calibrated using a leveling tool. A 30 μL aliquot of MAPbX₃ precursor solution was then deposited onto the substrate ahead of the auxiliary blade, which was positioned approximately 10 μm above the substrate to regulate the height of the precursor solution entering the PDMS template. By lowering the blade, the PDMS mold was brought into contact with the substrate surface. Capillary forces ensured rapid filling of the confined space as the precursor solution reached the microchannel openings. Note that achieving proper contact between PDMS and the substrate is the critical point of this strategy. Our standard procedure involves placing a precisely measured 50 μm thick flat metal foil (e.g., aluminum foil) on the substrate, then pressing down the PDMS block until the foil can be slightly slid out but not completely clamped (indicating contact without significant deformation). After removing the foil, PDMS elastic recovery bridges the 50 μm gap to achieve soft contact. This technique only provides a main idea, because it is difficult to ensure that the mechanical properties of the PDMS template actually produced are the same, so we can make fine adjustments on this basis. The blade was then moved steadily at an optimized speed of 0.08 mm s⁻¹, while maintaining the substrate at a temperature of 80 °C. Due to differences in wettability, the SCNWAs adhered to the substrate surface, with their coverage area determined by the volume of the precursor solution. Following the coating process, the substrate was immediately transferred to a hot plate and annealed at 100 °C for 2 min. This method successfully produced SCNWAs' active layers exceeding 12 cm² in area, with all experiments conducted under ambient conditions. Lateral-type photoresistors were fabricated by depositing interdigital Au electrodes (100 nm thickness) onto the arrays via vacuum evaporation using a metal mask, with an inter-electrode gap of approximately 35 μm. The active area of the device is about $35 \times 10 \times 10$ μm², with 10 nanowires covered by a pair of electrodes (Supplementary Fig. 22).

## Computational method

First-principles calculations were carried out within the framework of density functional theory (DFT) using the projector augmented-wave formalism, as implemented in the Vienna ab initio simulation package. The exchange–correlation interactions were described by the generalized gradient approximation parameterized by Perdew, Burke, and Ernzerhof. To account for long-range dispersion forces, the DFT-D3 correction scheme was applied. A kinetic energy cutoff of 400 eV was employed for the plane-wave basis set, and the electronic self-consistency criterion was set to $10^{-5}$ eV. To avoid spurious interactions between periodic images, a vacuum slab of 18 Å was introduced along the out-of-plane direction. Brillouin zone sampling was performed using

a $2 \times 2 \times 1$ Monkhorst–Pack k-point grid. Structural relaxations were carried out until the residual atomic forces were reduced below 0.03 eV Å⁻¹.

## Characterizations and photodetector measurement

The morphology of the perovskite SCNWAs was examined using an optical microscope (OM, KEYENCE VHX-500F), an IMA-hyperspectral microscope (Photon, Japan), and a scanning electron microscope (SEM, JEOL JSM-7610F) operated at an accelerating voltage of 5 kV. The crystalline structure and elemental composition of the thin films were further investigated by transmission electron microscopy (TEM, JEOL JEM-2100F) in conjunction with energy-dispersive X-ray spectroscopy (EDS, AZtec software, Oxford Instruments). TEM specimens were prepared with a cryo-focused ion beam system (Cryo-FIB, Thermo Fisher Scientific Helios Nanolab G3 CX). Large-area crystallinity was evaluated using X-ray diffraction (XRD, Panalytical XPert) with Cu $K\alpha$ radiation ($\lambda = 1.54178$ Å). Pole figure measurements were performed on a Rigaku SmartLab diffractometer equipped with a rotating copper anode operated at 45 kV and 160 mA (7.2 kW). The incident X-rays were monochromatized to 1.5406 Å using a Johansson monochromator and parallelized with a Goebel mirror. A divergence of 0.5° in the in-plane direction was introduced via Soller slits. The incident beam was collimated to 1.0 mm × 0.5 mm, while the scattered beam was further collimated with 0.5° Soller slits. A HyPix 300 detector was employed to record the diffraction patterns. The <001> and <101> reflections of MAPbBr₃ were detected at scattering angles of $2\theta = 14.93°$ and $2\theta = 26.64°$, respectively. Pole figures were obtained by $\beta$ scans with a step size of 1° and a scan speed of 100° s⁻¹. The inclination angle ($\alpha$) was varied in increments of 1° during successive scans. Steady-state photoluminescence (PL) spectra were acquired in a backscattering geometry using a 375 nm excitation laser. The emitted light was dispersed by an iHR320 monochromator (Horiba Jobin-Yvon) and detected with a Peltier-cooled Si CCD (Synapse, Horiba Jobin-Yvon). Transient PL dynamics were probed with a 402 nm excitation source and analyzed on a FluoTime 300 spectrometer. X-ray photoelectron spectroscopy (XPS) measurements were performed with an ESCALab 220i-XL system (VG Scientific) under 300 W Al Kα irradiation. Depth profiling was conducted using a VG EX 05 mini-beam ion gun with 1000 eV Ar⁺ ions and a 400 μm beam diameter. The etching rate was maintained at approximately 0.1 nm s⁻¹, calibrated against a SiO₂/Si reference sample. Fourier-transform infrared (FTIR) spectra were recorded with a Nicolet iS5 instrument (Thermo Fisher). Surface wettability was evaluated through contact angle measurements using a Dataphysics OCA 20 system. The absolute photoluminescence quantum yield (PLQY) was determined following the method described by de Mello et al.[54]. Samples were placed inside a 30 cm Spectralon-coated integrating sphere coupled to an AvaSpec-2048L spectrometer via an optical fiber. Optical absorption spectra of MAPbBr₃ SCNWAs were measured at room temperature using a PerkinElmer Lambda 950 UV–vis–NIR spectrophotometer equipped with an integrating sphere, with the scanning range set from 450 to 650 nm. The spectra were recorded with an excitation wavelength of 405 nm and corrected for the spectral sensitivity of the setup, determined with the help of a calibrated Xe lamp (Hamamatsu L7810-02). Thermal admittance spectroscopy (TAS) was performed using a ZAHNER PP211 covering a frequency range from 0.1 to 10⁶ Hz in the dark. All SCLC measurements were performed on hole-only devices with the structure Au/SCNWAs/Au, where Au electrodes (150 nm) were thermally evaporated onto both ends of the SCNWAs with an interelectrode spacing of approximately 35 μm. $I–V$ curves were recorded using a Keysight B2901A source meter. To minimize the influence of ion migration, we adopted a scan rate of 10 V s⁻¹, which showed negligible hysteresis. For stability evaluation, five identical devices were simultaneously aged in an artificial weathering chamber (LHL-114, Espec Corp.) under controlled conditions (25 °C, 85% RH). The measured parameters were averaged across all samples at each predetermined time interval. For stability evaluation, five identical devices were simultaneously aged in

an artificial weathering chamber (LHL-114, Espec Corp.) under controlled conditions (25 °C, 85% RH). The measured parameters were averaged across all samples at each predetermined time interval. The photoluminescence spectra were obtained using a 405 nm excitation source, and the data were corrected for the spectral response of the system with reference to a calibrated Xe lamp (Hamamatsu L7810-02). Thermal admittance spectroscopy (TAS) was conducted in the dark with a ZAHNER PP211 analyzer, covering a frequency range from 0.1 to $10^6$ Hz. Space-charge-limited current (SCLC) measurements were carried out on hole-only devices with the configuration Au/SCNWAs/Au, in which Au electrodes (150 nm) were thermally deposited at both ends of the SCNWAs, separated by approximately 35 μm. Device stability was evaluated using five identical samples, aged simultaneously in an environmental chamber (LHL-114, Espec Corp.) under controlled conditions of 25 °C and 85% relative humidity. The measured parameters were averaged across all devices at each designated time interval. Current–voltage ($I$–$V$) characteristics were measured using a semiconductor characterization system (Keithley 4200) integrated with a probe station (NKT Photonics) and a computer-controlled analog-to-digital converter. The incident light intensity was calibrated with an optical power meter (Newport, Model 1918-R). Temporal photoresponse measurements were conducted with a precision source meter (Keysight B2901A). The photocurrent spectra were obtained using a CIMPS-QE/IPCE testing setup equipped with a tunable light source (TLS03, ZAHNER). Noise current measurements of the photodetectors were performed in the dark using a lock-in amplifier (SR830). To evaluate imaging capability, a 2 mm thick stainless steel mask was used to project the optical pattern IMEET onto a 10×50 photodetector array. The projected image was subsequently converted into an electrical signal by recording the current from each pixel.

## Data availability

All the main data are available in the main text, the Supplementary Information, and the Source Data file. All other data of this study are available from the corresponding authors on request. Source data are provided with this paper.

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

## Acknowledgements

M.F., C.L., Z.X., Z.P., and J.T. are grateful for the financial support from the China Scholarship Council (CSC). J.B. and M.H.are grateful for the financial support from RTG2861-Planar carbon lattice (RTG2861) by Deutsche Forschungsgemeinschaft (DFG). M.F., C.L., H.H., Z.P., and J.T. gratefully acknowledge funding of the Erlangen Graduate School in Advanced Optical Technologies (SAOT) by the Bavarian State Ministry for Science and Art. C.J.B. and A.O. gratefully acknowledge financial support for this work by the DFG under GRK 2495/E. The authors acknowledge the financial support of the German Federal Ministry for Economic Affairs and Climate Action (Project Pero4PV, FKZ: 03EE1092A). K.G. and T.U. gratefully acknowledge funding by the consortium DAPHNE4NFDI in the context of the work of the NFDI e.V. The consortium is funded by the DFG Project No. 460248799.

## Author contributions

M.F. conceived the idea and designed the project. D.X., A.O., and C.J.B. supervised the research. M.F. fabricated perovskite single-crystal nanowire arrays and devices, conducted characterizations, analyzed the data, and wrote the first draft of the manuscript. J.B. and M.H. contributed to the photolithography process analysis and template provision. Z.L. contributed to the DFT simulations. Z.X. contributed to data collection from materials and devices, including electrochemical impedance and photodetector parameters. W.L. and X.W. assisted in XPS characterization and FTIR. K.G. and T.U. assisted in testing and analyzing the XRD polar figure data. Z.X., S.J., and J.E. measured SEM and EDS images. L.D. and T.W. contributed to FIB and TEM data collection and analysis. H.H. and J.T. performed PLQY measurements and participated in data discussions. C.L. and Z.P. assisted in conducting the variable light intensity TRPL experiments and data analysis. D.X., A.O., and C.J.B. revised the manuscript. All the authors revised and approved the manuscript.

## Funding

## Competing interests

The authors declare no competing interests.
