## [Transparent Peer Review file · Nature Communications]

Overcoming Size Limits with Dynamic Templates Enabling Large Area Single Crystal Nanowire Arrays for Photodetectors

Corresponding Author: Mr Mingjie Feng

Version 0:

Reviewer comments:

Reviewer #1

(Remarks to the Author)

The manuscript by Feng et al. presents a modified blade-coating method for preparing large-area perovskite single-crystal nanowire arrays (SCNWAs) to develop high-performance photodetectors. Unlike traditional template-based methods that use patterned substrates, this work employed a patterned PDMS template as the blade to grow highly ordered perovskite SCNWAs on diverse substrates, such as rigid silicon wafers and flexible polyimide substrates. Significantly, this dynamic template-assisted coating approach can eliminate the dependence on template size, thereby enabling the fabrication of perovskite SCNWAs with an area 12 times larger than the template itself. Furthermore, this work introduced a fluorinated passivating agent to passivate the surface defects of MAPbBr₃ single crystals, achieving photodetectors with a responsivity of 3320 A W⁻¹, a detectivity of 5.5×10^{14} Jones and an LDR of 160.3 dB. This work is likely to attract significant interest from the perovskite-based photodetector community. However, several issues need to be addressed before this manuscript can be considered for publication.

1. The discussion of experimental results currently lacks of clear organization and logical flow, occasionally deviating from conventional presentation methods. To improve clarity and coherence, please present the Results and Discussion sections sequentially, following the order of the figures.
2. The authors have emphasized that the dynamic template-assisted approach they developed can surmount the size limitations inherent in conventional blade-coating methods. However, the lateral dimensions of the fabricated perovskite single-crystalline nanowire arrays (SCNWAs) (Fig. 1b) remain limited to several centimeters. It remains to be explored whether this innovative method possesses the potential for further advancements in achieving larger lateral dimensions beyond the current scale.
3. Uniformity of crystal quality and optical properties at large scales is crucial for practical applications. Therefore, it is recommended to provide additional experimental results from multiple locations to demonstrate the uniformity of crystal morphology and optoelectronic properties across the large-area SCNWAs.
4. The DTA strategy exhibits good universality in the growth of MAPbX₃ (X=Cl, Br and I). Given that 2D perovskite materials with robust environmental stability are considered promising candidates for next-generation photodetectors (Nat. Electron. 2018, 1, 404 and Matter 2019, 1, 465), it would be of interest to further explore the feasibility of preparing 2D perovskite SCNWAs using the DTA method.
5. The labels for the XRD diffraction peaks in Supplementary Fig 10b are inconsistent with the description provided in the manuscript (Page 7-8). Typically, the (001) and (002) diffraction peaks for MAPbBr₃ single crystals are located at approximately 15° and 30°, respectively (Nat. Commun. 2020, 11, 1194 and Mater. Today 2019, 22, 67). However, in this manuscript, the (001) and (002) diffraction peaks for MAPbBr₃ SCNWAs exhibit a significant shift toward lower 2θ values. Why?
6. Numerous chemical passivation agents have been reported for passivating surface defects. Why did the author choose MTFA to modify the surface of MAPbBr₃ single crystal? Moreover, the motivation behind investigating the MTFA treatment is not clearly articulated in the manuscript. If similar investigations have been conducted previously, the authors should explain the necessity of this study and elucidate how it differs from prior works.
7. What does Supplementary Fig. 11-2 (Page 7 and Line 180) refer to?
8. The formula for calculating trap density on Page 10 and Line 271 has formatting issues. The authors should revise the formula and recalculate the trap density for perovskite SCNWAs.
9. On Page 10, the same concept of trap density is denoted as N_t in the SCLC measurement (Line 270), but as tDOS in TAS

measurement (Line 281). Please ensure consistent labelling of this concept throughout the manuscript.

10. The authors employ hole-only and electron-only devices to calculate the trap density of perovskite SCNWAs. To enhance the reproducibility and transparency of the study, the corresponding experimental details should be provided in the Experimental section.

11. On Page 11 and Line 299, the XRD patterns should be referenced as Fig. 3h instead of Fig. 3g. In addition, the measured R, LDR, D* and noise current at different biases are presented in Supplementary Fig. 18 instead of Supplementary Fig. 19. Please correct these references to ensure consistency with the figures provided.

12. The active area has a significant effect on the calculated responsivity and detectivity of photodetectors. Based on the description in the Experimental section and Supplementary Fig 18, the active area appears to be underestimated, which potentially leading to overestimated device parameters. To ensure transparency and accuracy, the corresponding calculation process should be detailed in the Supporting Information.

13. There are many formatting issues in this manuscript that affect the readability of the article. I list some examples below, and the authors need to check the manuscript carefully and repeatedly.

(i) On Page 6 and Line 157, "Landau-Levich regin" should be corrected "Landau-Levich regime".

(ii) The key terms that appear for the first time need to be abbreviated. However, the authors provide abbreviations at the end of the manuscript, such as SCNWAs (on Page 13, Line 327) and DFT (on Page 14, Line 389).

(iii) It is necessary to provide the correct superscripts and subscripts of relevant parameters. For example, superscripts in D* (Page 15, Line 393) and subscripts in lupper and llower (Page 13, Lines 341-342).

(iv) On Page 13, the responsivity for photodetector at 5 V and 10 V bias are 3320 A W⁻¹ (Line 336) and 4847 A/W (Line 355), respectively. In order to improve the standardization of the manuscript, it is recommended that the units be revised to a uniform format.

(v) Recheck all references for accuracy and completeness.

(vi) The description is inconsistent with the label in Fig. 3f.

Reviewer #2

(Remarks to the Author)

In this manuscript, Mingjie Feng and colleagues present a novel blade-coating-based process for the large-area growth of perovskite microwire arrays, successfully fabricating high-quality single-crystalline perovskite microwire arrays. Furthermore, the authors employed MTFA material to passivate defects within the single-crystalline microwire arrays, achieving an ultra-low defect state density of $5.37 \times 10^8 \text{ cm}^{-3}$. The resulting devices exhibit a remarkable responsivity of 3320 A/W and a specific detectivity of 5.5×10^{14} Jones, demonstrating superior optoelectronic performance compared to similar devices. However, the manuscript requires further refinement in several aspects, with certain sections needing more detailed explanations. Most critically, we question the reported defect density and optoelectronic performance, which may not be as exceptional as claimed. We suspect potential issues in the calculations of defect density and optoelectronic performance presented in the paper.

Therefore, we request that the authors address the following concerns thoroughly. The suitability of this manuscript for publication in Nature Communications will be evaluated based on the authors' responses to these issues.

1. On line 88, the authors introduce the abbreviation "R2R" for "roll-to-roll." However, this term appears only once in the main text. We recommend removing the abbreviation to maintain clarity and consistency.

2. On line 180, the authors reference Supplementary Fig. 11-2. However, this figure is not included in the supplementary information.

3. On lines 341–342, the authors refer to "where lupper and llower" without applying subscripts to the corresponding terms.

4. On line 353, the authors state, "Supplementary Fig. 18 shows the measured R, LDR,..." However, the corresponding figure in the supplementary information is actually Supplementary Fig. 19. We recommend that the authors carefully review the manuscript to ensure the accuracy of all figure references.

5. In Figure 1e, why does the non-microwire region exhibit a certain level of lead element presence, while the bromine element is distinctly more concentrated in the microwire region? The authors should provide a detailed explanation for this observation.

6. It is generally understood that the selective trapping of one type of carrier and the release of another at the surface of microwires/nanowires is a critical physical process enabling multiplied responsivity in one-dimensional devices.

Consequently, surface defects are considered a key factor in achieving high multiplication responses. However, the authors report maintaining exceptionally high responsivity despite passivating surface defects. The authors should clarify why such high responsivity persists post-passivation. Additionally, the manuscript does not present data comparing the device's photoresponsivity before and after MTFA passivation. We strongly recommend that the authors provide these data.

7. On line 214, the authors claim that binding energy calculations demonstrate that TFA preferentially adheres to the crystal surface. Could the authors clarify the methodology and evidence supporting this conclusion? A detailed explanation of how the binding energy calculations substantiate this preference is needed.

8. The experimental details require further elaboration. For instance, on line 228, the authors mention etching the microwire arrays. We recommend that the authors include a detailed description of the etching process in the experimental section to enhance the reproducibility and clarity of the study.

9. In Figure 3c, the authors present TRPL data for the microwire arrays but do not provide specific values for the carrier lifetime of the material before and after MTFA treatment. We recommend that the authors include these quantitative values to clearly demonstrate the impact of MTFA passivation.

10. In Supplementary Fig. 15, the authors present J-V curves (reverse and backward sweeps) for electron-only devices under different scan rates. However, several issues require clarification. First, the authors do not specify the detailed device structure of the electron-only devices. Second, the y-axis is labeled in current units, so the curve should be named I-V curve. Furthermore, the presented data clearly indicate a pronounced hysteresis effect in the devices, which is only partially

mitigated at higher scan rates. We recommend that the authors address these points by providing the device structure, correcting the name, and discussing the observed hysteresis in greater detail.

11. We have significant concerns regarding the authors' calculation of defect state density based on the SCLC theory. First, the formula presented on line 271 contains clear errors: (i) relevant terms lack appropriate subscripts, and (ii) the actual formula for calculating defect state density appears to be incorrect. The correct formula should be (Single-crystalline layered metal-halide perovskite nanowires for ultrasensitive photodetectors. *Nat Electron* 1, 404–410 (2018)). Furthermore, assuming a dielectric constant of approximately 20 for MAPbBr₃, and using the authors' provided electrode spacing of 25 μm and VTFL of 1.34 V, we estimate the defect density to be approximately $5 \times 10^{12} \text{ cm}^{-3}$. This value is significantly higher than the reported $8.72 \times 10^9 \text{ cm}^{-3}$. We request that the authors provide a detailed explanation for this discrepancy, including a thorough review of their calculation methodology and assumptions.

12. The effective area of the device is critical to the accuracy of the reported device performance metrics. In the Experimental Section, the authors state that the distance between device electrodes is approximately 25 μm. Based on the scale bar provided in Supplementary Fig. 18, we estimate the width of a single microwire to be approximately 10 μm. Consequently, the effective area of a single microwire in a device is approximately 250 μm². Given that each device contains 10 microwires, the total effective area of a device is approximately 2500 μm². However, the authors report a device area of $35 \times 5 \mu\text{m}^2$, which is 15 times smaller than the estimated effective area.

Could the authors clarify how the $35 \times 5 \mu\text{m}^2$ area was determined? If the device area has been miscalculated, the reported responsivity (R) and specific detectivity (D*) must be recalculated. We request a detailed explanation of the area calculation and, if necessary, a revision of the performance metrics based on the correct effective area.

Version 1:

Reviewer comments:

Reviewer #1

(Remarks to the Author)

The authors have addressed the questions raised by the reviewers. I have no further questions regarding this manuscript.

Reviewer #2

(Remarks to the Author)

The author has provided a detailed response to our earlier queries, addressing several initial concerns about the article. To further enhance the reliability of the reported specific detectivity (D*), we recommend including the instrumental baseline noise data in a relevant section of the paper. This addition would help confirm that the noise data presented in Figure 4d are attributable to the photodetector itself, rather than the instrumental noise background.

made.

Response to Reviewers' Comments and Revised Details Manuscript

ID: NCOMMS-25-27534-T

We greatly appreciate the reviewers' insightful comments and constructive suggestions, which have helped us to significantly improve the quality of our manuscript. We have carefully addressed all the concerns raised by the reviewers and hope that our responses meet their expectations.

The reviewers' comments are presented below in *italicized font*, with specific concerns numbered for clarity. Our responses are provided in **blue font**, and the corresponding revisions or additions to the manuscript are highlighted in **red**.

Reviewer #1:

“The manuscript by Feng et al. presents a modified blade-coating method for preparing large-area perovskite single-crystal nanowire arrays (SCNWAs) to develop high-performance photodetectors. Unlike traditional template-based methods that use patterned substrates, this work employed a patterned PDMS template as the blade to grow highly ordered perovskite SCNWAs on diverse substrates, such as rigid silicon wafers and flexible polyimide substrates. Significantly, this dynamic template-assisted coating approach can eliminate the dependence on template size, thereby enabling the fabrication of perovskite SCNWAs with an area 12 times larger than the template itself. Furthermore, this work introduced a fluorinated passivating agent to passivate the surface defects of MAPbBr₃ single crystals, achieving photodetectors with a responsivity of 3320 A W⁻¹, a detectivity of 5.5 × 10¹⁴ Jones and an LDR of 160.3 dB. This work is likely to attract significant interest from the perovskite-based photodetector community. However, several issues need to be addressed before this manuscript can be considered for publication.”

Response: We thank the reviewer for their positive feedback and for their constructive comments to improve our manuscript. Detailed responses to each point are provided below.

1. “The discussion of experimental results currently lacks of clear organization and logical flow, occasionally deviating from conventional presentation methods. To improve clarity and coherence, please present the Results and Discussion sections sequentially, following the order of the figures.”

Response: We sincerely appreciate the reviewer’s valuable suggestion to improve the clarity and logical structure of the *Results and Discussion* section. In response, we have carefully revised the manuscript to align the narrative flow with the actual figure sequence, enhancing overall readability and consistency.

In detail:

- In **Figure 1**, the original discussion order ($g \rightarrow f \rightarrow e$) has been revised to $e \rightarrow f \rightarrow g$, now matching the visual layout from left to right.
- In **Figure 2**, the sequence has been updated from $f \rightarrow e$ to $e \rightarrow f$.
- In **Figure 3**, we reorganized the discussion from $b \rightarrow c \rightarrow a$ to the more intuitive $a \rightarrow b \rightarrow c$ order.

We believe that this restructuring significantly improves the coherence of the experimental narrative and better guides the reader through the results. These changes have been implemented in the revised manuscript accordingly.

2. “The authors have emphasized that the dynamic template-assisted approach they developed can surmount the size limitations inherent in conventional blade-coating methods. However, the lateral dimensions of the fabricated perovskite single-crystalline nanowire arrays (SCNWAs) (Fig. 1b) remain limited to several centimeters. It remains to be explored whether this innovative method possesses the potential for further advancements in achieving larger lateral dimensions beyond the current scale.”

Response: We sincerely appreciate the reviewer’s insightful comments. We would like to clarify a potential misunderstanding regarding the size limitation mentioned in our manuscript. Specifically,

the size constraint we referred to does not stem from the conventional blade-coating technique itself, but rather from traditional template-assisted growth methods, where the lateral dimensions of the resulting nanostructures are strictly limited by the fixed physical size of the static template. In contrast, our dynamic template-assisted (DTA) method overcomes this limitation by allowing the template to move synchronously with the blade during the coating process. This enables continuous and scalable patterning of SCNWAs along the coating direction, achieving **much longer longitudinal lengths** than those attainable by previous static-template-based strategies.

We understand the reviewer's suggestion pertains to whether our method can also achieve **larger lateral widths**. Currently, the lateral (cross-blade) width of the SCNWAs is ~2 cm, which represents the largest width we could achieve with consistent alignment and crystal continuity across the array. In our experiments, further increasing the lateral size of the template often led to discontinuities or breaks in the SCNWAs. We attribute this to the increased likelihood of **nonuniform contact** between the enlarged template and the substrate, which disrupts the confined crystal growth conditions necessary for maintaining wire integrity.

To further illustrate the lateral scalability of our DTA strategy, **Fig. R1** presents both the working principle and practical demonstration of multi-pass printing. As shown in **Fig. R1a**, the template is sequentially repositioned along the lateral direction after each coating step, enabling the fabrication of parallel MAPbBr₃ SCNWAs over an extended lateral width. **Fig. R1b** shows a photograph of the resulting wafer-scale substrate after three consecutive coatings, clearly demonstrating the formation of large-area patterned SCNWAs. Notably, as shown in **Fig. R1c**, the printed regions exhibit pronounced optical diffraction (grating) effects under ambient light, serving as visual evidence of the highly ordered periodic structure (*Adv. Mater.* 2020, 32, 2001998; *Adv. Mater.* 2023, 2310427). Each nanowire within the array exhibits a uniform width of approximately 10 μm, with an interwire spacing of 8 μm, confirming the formation of well-aligned MAPbBr₃ SCNWAs with single-crystalline order. At this proof-of-concept stage, the spacing between adjacent arrays is approximately 0.3 cm, maintained using PI tape to prevent the precursor solution from spilling or redissolving previously formed structures. We believe that SCNWAs based on this approach are dimensionally unlimited in the lateral direction through multiple coatings.

Fig. R 1 (a) Schematic illustration of the dynamic template-assisted blade-coating process for large-area MAPbBr₃ SCNWAs. The template synchronously moves with the blade across the substrate, enabling sequential deposition in adjacent regions. (b) Optical photograph of a 4-inch Si wafer with three adjacent SCNWAs regions fabricated via consecutive horizontal blade passes. (c) Oblique-angle optical image of the patterned MAPbBr₃ SCNWAs.

Revised Statement:

- We have added Fig. R1 to the Supplementary Information, where it can be seen in Supplementary Fig. 9.
- The following content: “We further explored the potential of the DTA strategy for lateral dimensional scaling by sequentially shifting the substrate and repeating the coating process. This approach enabled the fabrication of multiple adjacent SCNWAs regions on a 4-inch wafer, as demonstrated in Supplementary Fig. 9.” was added to the main text on Page 6, Line 160.

3. *“Uniformity of crystal quality and optical properties at large scales is crucial for practical applications. Therefore, it is recommended to provide additional experimental results from multiple locations to demonstrate the uniformity of crystal morphology and optoelectronic properties across the large-area SCNWAs.”*

Response: We sincerely thank the reviewer for highlighting the importance of assessing the uniformity of crystal morphology and optoelectronic properties over large-scale areas. In fact, as the reviewer is concerned, uniformity is a key prerequisite for the practical application of large-area SCNWAs as it ensures the consistency, reproducibility of device performance, and the scalability of the manufacturing process.

To comprehensively evaluate and demonstrate the uniformity of our MAPbBr₃ SCNWAs, we have carried out additional experiments from optical, electrical, and morphological perspectives:

- **Optical uniformity:** As shown in **Fig. R2a**, we selected 43 different measurement points distributed across the entire ~20 mm × 60 mm SCNWAs area. At each point, we measured the photoluminescence (PL) spectra and analyzed the peak positions and full widths at half maximum (FWHM). The results are presented in **Fig. R2b** as a 3D pseudo color plot showing that the emission peak wavelength remains nearly constant across all 43 measured locations, indicating good optical uniformity. This is further supported by the statistical analysis summarized in Table R1, which shows a standard deviation of only 1.4% for the peak position and 2.4% for the full width at half FWHM. Additionally, **Fig. R2c** presents a three-dimensional intensity plot of the PL spectra from the same 43 positions, revealing minimal variation in PL intensity and spectral shape across the scanned region, further confirming the uniform optoelectronic properties of the large-area MAPbBr₃ SCNWAs (*Adv. Mater.* 2018, 30, 1707314).
- **Dynamic optical imaging:** To further visualize the uniformity and continuity of the SCNWAs, we recorded a video during lateral movement along the nanowire array. This was performed using an IMA-hyperspectral microscope system combined with a BRESSER MikroCamII 20 MP 1" microscope camera. By continuously moving the microscope stage along the nanowire alignment direction, with a maximum travel distance of approximately 50 mm and each nanowire measuring about 5 μm in width, we recorded the morphology and structural uniformity of the SCNWAs. The video clearly demonstrates the uninterrupted and homogeneous nature of the single-crystal nanowire arrays over macroscopic lengths (**Supplementary Movie 1**).
- **Electrical uniformity:** As electrical properties are equally important for device performance, we first analyzed the dark current distribution from 50 randomly selected positions in the original manuscript (Fig. 4f). Furthermore, we now provide measurements of transient photoresponse and decay dynamics from 43 additional points, covering the same 20 mm × 60 mm region. The data are summarized in Table R1, which confirms consistent photoelectric behavior throughout the large-area SCNWAs, with an average rise time of 102.2 μs and a standard deviation of 3.7%, as well as an average decay time of 64.7 μs with a standard deviation of 4.0%.

In summary, these results demonstrate that the single-crystal nanowire arrays we fabricated exhibit

generally good uniformity in crystal quality and optoelectronic properties across a large area. However, a certain degree of variation still exists. We attribute this mainly to edge effects occurring at the beginning and end of the blade-coating process, where the crystal growth conditions may not be as uniform as in the central region. Addressing these edge-related non-uniformities will be an important focus of our future optimization efforts.

Fig. R2 (a) Schematic illustration of the large-area MAPbBr₃ SCNWAs sample, showing the distribution of 43 measurement points (red dots) across a ~ 20 mm \times 60 mm region selected for PL and transient photoresponse testing. (b) 3D pseudocolor plot of the evolution of the PL spectra collected from different points along the SCNWAs. (c) Three-dimensional intensity plot of the PL spectra measured across the same 43 locations, revealing minimal variation in spectral intensity and shape.

Table R1. PL peak position, FWHM, and transient response of large-area MAPbBr₃ SCNWAs grown via the DTA strategy, measured at 43 different positions.

Locations	Peak position (nm)	FWHM (nm)	τ_{rise} (μs)	τ_{decay} (μs)
1	543	26.7	112.7	74.3
2	541	23.8	101.2	67.5
3	540	21.3	103.4	64.2
4	542	20.6	105.7	61.1
5	541	24.8	108.6	63.5
6	541	23.7	101.0	64.9
7	541	26.7	107.3	62.8
8	541	26.4	112.5	63.7
9	541	26.1	99.3	65.7
10	541	24.4	101.6	71.9
11	541	25.7	100.7	64.7
12	543	24.6	101.5	70.1

13	542	23.2	110.8	66.2
14	542	24.2	100.2	62.5
15	540	28.1	101.4	69.6
16	541	21.6	106.5	62.4
17	541	21.5	100.9	64.6
18	541	22.7	101.2	62.8
19	541	22.7	102.3	64.6
20	541	21.9	106.1	58.7
21	542	25.4	104.6	63.3
22	541	26.4	99.7	60.8
23	540	21.3	102.1	61.5
24	540	20.8	101.7	64.9
25	540	24.6	99.5	67.6
26	538	25.3	100.3	59.3
27	539	22.6	99.5	60.7
28	539	25.7	98.2	71.3
29	538	22.3	99.7	64.7
30	540	20.9	99.1	70.6
31	539	20.4	98.9	64.3
32	541	23.1	101.3	65.7
33	541	21.2	99.9	59.6
34	542	30.4	99.0	57.4
35	539	29.2	99.3	58.8
36	538	25.0	99.5	61.7
37	538	21.6	101.3	63.7
38	538	21.5	100.6	71.9
39	538	21.7	99.8	65.4
40	539	22.4	99.4	62.3
41	541	24.6	99.4	66.3
42	538	22.7	99.5	70.7
43	541	22.2	105.9	66.1
Mean	540.3	23.8	102.2	64.7
Standard Deviation	1.4	2.4	3.7	4.0

Revised Statement:

- We have added Fig. R2, Table R1, and Supplementary Movie 1 to the Supplementary Information, where they can be seen in Supplementary Fig. 19, Supplementary Table 5, and

Supplementary Movie 1.

- The following content: “Demonstrating the spatial uniformity of crystal quality and optoelectronic properties is essential for validating the scalability and practical applicability of large-area SCNWAs. Spatially resolved photoluminescence (PL) and transient response measurements were therefore performed at 43 locations across an area of approximately 20 mm × 60 mm. The 3D pseudocolor PL mapping (Supplementary Fig. 19b) shows minimal peak shift (standard deviation: 1.4%), while the corresponding intensity plot (Supplementary Fig. 19c) reveals consistent spectral shapes (FWHM variation: 2.4%, Supplementary Table 5). Supplementary Movie 1 further confirms the structural consistency over a 50 mm serial scan. Additionally, transient photoresponse data exhibit uniformly stable rise and decay behaviors across the sampled positions, highlighting the performance and spatial uniformity of the SCNWAs.” was added to the main text on Page 12, Line 322.

4. *“The DTA strategy exhibits good universality in the growth of MAPbX₃ (X=Cl, Br, and I). Given that 2D perovskite materials with robust environmental stability are considered promising candidates for next-generation photodetectors (Nat. Electron. 2018, 1, 404 and Matter 2019, 1, 465), it would be of interest to further explore the feasibility of preparing 2D perovskite SCNWAs using the DTA method.”*

Response: We sincerely thank the reviewer for the insightful suggestion regarding the applicability of the DTA strategy to 2D perovskite systems. As the reviewer correctly pointed out, 2D perovskites—featuring robust chemical stability, tunable optoelectronic characteristics, and intrinsic multiple quantum well structures—have emerged as one of the most promising candidates for next-generation electronic and optoelectronic devices. In response, we have extended our method to fabricate SCNWAs based on 2D perovskite compositions, including PEA₂PbI₄ and BA₂MA_{n-1}Pb_nBr_{3n+1} with n = 1, 2, and 4.

(1) Demonstration of DTA Compatibility with 2D Perovskite Systems:

- The precursor solution formulation was primarily based on previous reports, with a key modification: we consistently used DMSO as the solvent across all systems to slow solvent evaporation and control nucleation kinetics (*Adv. Funct. Mater.* 2024, 2418968; *Nanoscale*,

2019,11, 18272).

- Based on iterative experimental feedback, we optimized two main parameters—precursor concentration and coating speed—while keeping the substrate temperature constant at 70 °C. For 2D perovskites such as PEA_2PbI_4 and BA_2PbBr_4 , a concentration of 0.3 M and a coating speed of 60 $\mu\text{m/s}$ were used. For quasi-2D systems, the concentration was increased to 0.6 M with a speed of 100 $\mu\text{m/s}$. We observed that as the value of n increases, the crystallization rate tends to decrease, necessitating tuning of experimental parameters to accelerate solvent evaporation and maintain a favorable balance between nucleation and crystal growth.
- As shown in Fig. R3, our results demonstrate that the DTA strategy can be effectively extended to a variety of 2D and quasi-2D perovskite systems, including BA_2PbBr_4 ($n = 1$), $\text{BA}_2\text{MAPb}_2\text{Br}_7$ ($n = 2$), $\text{BA}_2\text{MA}_3\text{Pb}_4\text{Br}_{13}$ ($n = 4$), and PEA_2PbI_4 ($n = 1$). The resulting SCNWAs exhibit good film continuity and uniformity over centimeter-scale dimensions. Notably, the iridescent patterns observed from different viewing angles—particularly in panels (b) and (d)—further confirm the formation of well-aligned and uniformly distributed nanowire grating structures.

Fig. R3 Photographs of large-area 2D perovskite SCNWAs prepared using the DTA method. (a) Mixed- n phase bromide systems: $\text{BA}_2\text{MA}_3\text{Pb}_4\text{Br}_{13}$ ($n = 4$), BA_2PbBr_4 ($n = 1$), and $\text{BA}_2\text{MAPb}_2\text{Br}_7$ ($n = 2$). (c) Pure $n = 1$ iodide system: PEA_2PbI_4 . The iridescent patterns, especially visible in (b) and (d) from different viewing angles.

(2) Structural Morphology and Crystallinity Assessment of 2D SCNWAs

To further verify the structural integrity and crystallinity of the fabricated 2D perovskite SCNWAs,

we conducted optical microscopy imaging and X-ray diffraction (XRD) characterization on representative samples.

As shown in Fig. R4a-d, all four types of 2D SCNWAs—including $(\text{PEA})_2\text{PbI}_4$, $(\text{BA})_2\text{PbBr}_4$, $(\text{BA})_2\text{MAPb}_2\text{Br}_7$, and $(\text{BA})_2\text{MA}_3\text{Pb}_4\text{Br}_{13}$ —exhibit highly periodic and well-aligned nanowire arrays over large areas. The arrays demonstrate good uniformity and parallelism, indicative of controlled nucleation and directional crystallization enabled by the DTA process. The corresponding XRD patterns (Fig. R4e-h) show sharp diffraction peaks at low angles, which are characteristic of layered 2D perovskite structures (*Nat. Commun.* 2023, 14, 2808; *Adv. Mater.* 2020, 32, 1907364; *Adv. Funct. Mater.* 2024, 2418968). For the $(\text{PEA})_2\text{PbI}_4$ and $(\text{BA})_2\text{PbBr}_4$ samples, the peaks correspond well to the (001) reflections, indicating that the crystal planes are preferentially oriented perpendicular to the substrate surface. As the n value increases in the $\text{BA}_2\text{MA}_{n-1}\text{Pb}_n\text{Br}_{3n+1}$ systems ($n = 2, 4$), additional peaks appear, reflecting the increased complexity and interlayer spacing variation in quasi-2D phases. As the value of n increases in the $\text{BA}_2\text{MA}_{n-1}\text{Pb}_n\text{Br}_{3n+1}$ systems (particularly $n = 4$), additional diffraction peaks emerge, indicating increased structural complexity and phase diversity. Notably, for the $n = 4$ sample ($(\text{BA})_2\text{MA}_3\text{Pb}_4\text{Br}_{13}$), distinct peaks near 15° and 30° correspond well to the (001) and (002) planes of the 3D-like MAPbBr_3 phase. This observation suggests partial phase segregation or the coexistence of quasi-2D and 3D-like domains. Such behavior is commonly reported in high- n 2D perovskite systems, where incomplete incorporation of the bulky spacer cation (BA^+) and relatively lower spacer-to-framework ratio promote the formation of intermediate or even 3D perovskite components (*Adv. Funct. Mater.* 2023, 33, 2209249). These results collectively validate the compatibility of the DTA strategy with 2D and quasi-2D perovskites and underscore its ability to produce highly ordered, single-crystalline nanowire arrays.

Fig. R4 Optical microscopy images (a–d) and corresponding XRD patterns (e–h) of 2D and quasi-2D perovskite SCNWAs prepared via the DTA method. (a, e) $(\text{PEA})_2\text{PbI}_4$ ($n = 1$), (b, f) $(\text{BA})_2\text{PbBr}_4$ ($n = 1$), (c, g) $\text{BA}_2\text{MAPb}_2\text{Br}_7$ ($n = 2$), and (d, h) $\text{BA}_2\text{MA}_3\text{Pb}_4\text{Br}_{13}$ ($n = 4$).

5. “The labels for the XRD diffraction peaks in Supplementary Fig 10b are inconsistent with the description provided in the manuscript (Page 7-8). Typically, the (001) and (002) diffraction peaks for MAPbBr_3 single crystals are located at approximately 15° and 30° , respectively (*Nat. Commun.* 2020, 11, 1194 and *Mater. Today* 2019, 22, 67). However, in this manuscript, the (001) and (002) diffraction peaks for MAPbBr_3 SCNWAs exhibit a significant shift toward lower 2θ values. Why?”

Response: We sincerely thank the Reviewer for pointing out this important issue. Upon careful re-examination, we realized that due to a clerical error, the XRD pattern shown in Supplementary Fig. 10b was mistakenly taken from MAPbI_3 SCNWAs rather than MAPbBr_3 SCNWAs. The diffraction peaks near $\sim 14.2^\circ$ and $\sim 28.5^\circ$ observed in the original figure correspond well to the (110) and (220) planes of MAPbI_3 , as reported in previous literature (*Adv. Mater.* 2020, 32, 1908340; *Adv. Funct. Mater.* 2022, 32, 2112758; *J. Mater. Sci. - Mater. Electron.* 2022, 33, 21531). This further confirms that the figure was incorrectly inserted, and the shift in peak positions was not due to changes in the MAPbBr_3 lattice or material properties.

We have now corrected this mistake by replacing the XRD data in Supplementary Fig. 11b with the appropriate pattern for MAPbBr_3 SCNWAs. The updated Fig. R5 clearly shows the (001) and (002) diffraction peaks at approximately 15° and 30° , in agreement with previously reported values (*Nat. Commun.* 2020, 11, 1194; *Mater. Today.* 2019, 22, 67). The corresponding description in the main text has also been revised to ensure consistency. We sincerely apologize for this mistake and

appreciate the Reviewer's careful reading.

Fig. R5 XRD pattern of the MAPbBr₃ thin films fabricated with and without using the DTA strategy.

Revised Statement:

- Fig. R5 is used to replace Supplementary Fig. 11b

6. *“Numerous chemical passivation agents have been reported for passivating surface defects. Why did the author choose MTEFA to modify the surface of MAPbBr₃ single crystal? Moreover, the motivation behind investigating the MTEFA treatment is not clearly articulated in the manuscript. If similar investigations have been conducted previously, the authors should explain the necessity of this study and elucidate how it differs from prior works.”*

Response: We appreciate the reviewer's thoughtful question regarding the motivation for choosing MTEFA and the novelty of our study.

(1) Why surface passivation of MAPbBr₃ SCNWAs is necessary:

Our rationale for surface passivation considers two key aspects:

- It is well known that solution-processed perovskite materials inevitably suffer from defect formation due to incomplete crystallization and dynamic ion migration. While hybrid perovskite single crystals exhibit significantly reduced bulk defect densities compared to polycrystalline films, surface defects—particularly undercoordinated Pb²⁺ ions, halide vacancies, Pb–Pb dimers, and other related surface states—persist and play a non-negligible role in carrier trapping and nonradiative recombination (*Nat. Photon.* 2024, 18, 250; *Nat. Rev. Mater.* 2020, 5, 809; *J. Am. Chem. Soc.* 2018, 140, 46, 15753). These surface-related imperfections are especially detrimental in photodetector applications, where efficient charge transport and fast

photoresponse are required. Therefore, implementing effective surface passivation strategies is essential to fully harness the intrinsic optoelectronic quality of MAPbBr₃ single crystals and further boost their device-level performance.

- In addition to suppressing nonradiative surface recombination, surface passivation is particularly crucial for MAPbBr₃ SCNWAs due to their inherently large surface-to-volume ratio. The extensive exposure of crystal surfaces to ambient atmosphere increases the likelihood of moisture infiltration, ion migration, and chemical degradation (*Adv. Mater.* 2020, 32, 2001998; *Nano Lett.* 2016, 16, 7446). These effects are further intensified in lateral device architectures where the active region lies directly at the exposed surface. Therefore, introducing a suitable passivation layer not only reduces surface trap density and suppresses nonradiative recombination but also acts as a protective barrier, enhancing both the charge transport efficiency and the environmental stability of the nanowire-based photodetectors.

(2) Why we chose MTFA as the passivation agent:

Our selection of MTFA is based on four key considerations:

- First, considering the possible defect species that exist on the surface of MAPbBr₃ SCNWAs—including undercoordinated Pb²⁺ ions, Pb–Pb dimers, and MA⁺ vacancies—it is important to introduce a passivation molecule capable of multi-site and multifunctional coordination. Such a passivating agent should ideally bind with undercoordinated Pb²⁺ to suppress both shallow and deep-level traps, while simultaneously interacting with organic A-site species to reduce vacancy concentrations. In this context, methylamine trifluoroacetate (MTFA) is a particularly suitable candidate. Its trifluoroacetate (TFA⁻) anion contains both a carbonyl (C=O) group, which can coordinate with Pb²⁺/Pb⁰ species (*Angew. Chem. Int. Ed.* 2025, 64, e202420369), and strongly electronegative F atoms that can form hydrogen bonds with the –NH₃⁺ group of MA⁺ cations. Therefore, the passivation effect should be significantly superior to other commonly used passivating groups, such as thiocyanate (SCN⁻), acetate (Ac⁻), and formate (HCOO⁻), etc (*Joule.* 2024, 8, 2283).
- Second, MAPbBr₃ SCNWAs have a high surface-area-to-volume ratio, making them more prone to environmental degradation. Besides the electronegativity and coordination capabilities of F, another key characteristic is its high hydrophobicity. The surface-bound fluorine atoms in TFA⁻ form a hydrophobic layer that helps resist moisture ingress and improves long-term device

stability (*Nat. Energy*. 2019, 4, 408).

- The observed hydrophobicity enhancement upon introducing MTFA can be attributed to the unique chemical structure of its anionic component, TFA⁻. This anion contains a highly electronegative trifluoromethyl group (-CF₃), which exhibits extremely low polarizability and surface energy. The strong C-F bonds in -CF₃ groups are known to be chemically inert, non-polar, and hydrophobic, leading to poor affinity with substrate (*Chem. Commun.* 2019, 55, 11059; *Macromolecules*. 2024, 57, 5189; *J. Appl. Polym. Sci.* 2022, 139, e52132). When MTFA is incorporated into the precursor solution, the -CF₃ groups preferentially orient toward the air-liquid interface, thereby reducing the overall surface energy of the droplet. In our measurements, the contact angle of the MAPbBr₃ precursor solution was found to gradually increase with increasing MTFA concentration, confirming the enhanced hydrophobic character imparted by the -CF₃ moieties at the liquid interface (Fig. R6). The hydrophobic nature of MTFA-modified precursors not only alters the wetting dynamics at the substrate interface but also enhances in-plane precursor ion diffusion by suppressing solvent-substrate interactions (*Langmuir*, 2009, 25, 10768). This improved ion mobility promotes more uniform and continuous crystal growth. Such a strategy has been widely employed to facilitate the formation of large-grained polycrystalline perovskite films, as well as to enable space-confinement-assisted growth of high-quality, large-area perovskite single crystals (*Nat. Photon.* 2023, 17, 401; *Nat. Commun.* 2017, 8, 1890; *Nat. Commun.* 2015, 6, 7747).
- MTFA, serving as a passivating agent, can be incorporated directly into the perovskite precursor solution, thereby circumventing the need for secondary deposition steps, which can be particularly advantageous when fabricating large-area devices.

(3) Novelty relative to previous studies:

Our implementation of MTFA is driven by three key innovations:

- To date, we are only aware of one prior report by Liu et al., in which MTFA was incorporated into MAPbI₃ precursor solutions to promote the formation of large-grained, stable perovskite thin films (*Chem. Commun.* 2019, 55, 11059). Nevertheless, the application of MTFA or structurally similar species to MAPbBr₃ single crystals, especially in low-dimensional geometries, has been scarcely investigated. Our study thus seeks to expand the utility of TFA⁻-based molecular engineering into high-quality single-crystal systems and assess its impact on

both optoelectronic performance and long-term stability.

- In addition, we systematically investigated the influence of MTFA concentration on the DTA growth of MAPbBr₃ SCNWAs using a blade-coating-based confined crystallization strategy. Through this study, we identified an optimal MTFA-to-MAPbBr₃ mass ratio of 1:20 (Supplementary Fig. 18), at which the resulting SCNWAs exhibit superior structural uniformity, reduced trap density, and enhanced device performance. When the MTFA concentration is below this threshold, excessive interaction between solutes and the substrate impedes solute transport, limiting the formation of continuous, large-area nanowires. In contrast, when the MTFA content is too high, the precursor solution becomes overly hydrophobic, significantly weakening substrate wetting and disrupting the confined solvent flow and crystallization front stability. This imbalance adversely affects crystallization kinetics and compromises film integrity. This concentration-optimization study not only deepens our understanding of additive-mediated crystal growth in confined geometries but also provides practical guidance for scalable fabrication of high-quality perovskite micro/nanostructures.
- Moreover, in our study, we not only assessed the passivation effect of MTFA through comprehensive experimental characterization, but also employed DFT calculations to gain mechanistic insights into how MTFA functions as a passivating agent—specifically, how it effectively reduces defect densities in MAPbBr₃ SCNWAs and simultaneously enhances their long-term stability.

Fig. R6. Contact angle images of MAPbBr₃ precursor droplets on Si substrates with increasing MTFA to MAPbBr₃ weight ratios: 0, 1:30, 1:25, 1:20, 1:15, and 1:10.

Revised Statement:

- We have incorporated Fig. R6 into Supplementary Fig. 18 to illustrate the effect of MTFA on the

wettability of the precursor solution on the substrate surface.

- As rightly pointed out by the reviewer, the motivation for employing MTFA in the surface modification of MAPbBr₃ SCNWAs was not clearly articulated in the main text. To address this, the following content: “To achieve effective passivation, we introduced methylammonium trifluoroacetate (MTFA) into the precursor solution. Density functional theory (DFT) calculations reveal that the carboxylate group (O=C–O[–]) of trifluoroacetate (TFA[–]) acts as a strong electron-donating Lewis base due to its high electronegativity (Fig. 2a). As illustrated in Fig. 2b, TFA[–] coordinates with unsaturated Pb²⁺ ions, stabilizing Pb sites and suppressing Pb dimer formation. Additionally, fluorine atoms in the –CF₃ group form hydrogen bonds with MA⁺ cations, stabilizing MA sites and reducing V_{MA} density (Supplementary Fig. 13).” ***was replaced by*** “To mitigate the typical surface defects in MAPbBr₃ SCNWAs—including undercoordinated Pb²⁺ ions, Pb–Pb dimers, and MA⁺ vacancies—we sought a passivation agent capable of simultaneously interacting with both metal-related and A-site-related defect species. MTFA was selected due to its unique dual-functionality: TFA[–] anion contains a Lewis-basic carboxylate group and highly electronegative –CF₃ groups, which can engage in complementary interactions with different types of defects. Even though this type of passivating agent—as well as related TFA[–]-based molecules—has been successfully employed to improve crystallinity and stability in polycrystalline perovskite films, its application to single-crystalline materials, particularly 1D-SCNWAs, remains largely unexplored, yet appears to involve an intriguing and potentially significant underlying mechanism. To bridge this gap, we incorporated MTFA into the precursor solution. Density functional theory (DFT) calculations reveal that the carboxylate group (O=C–O[–]) of TFA[–] acts as a strong electron-donating Lewis base due to its high electronegativity (Fig. 2a). As illustrated in Fig. 2b, TFA[–] coordinates with undercoordinated Pb²⁺ ions, stabilizing Pb sites and suppressing Pb dimer formation. Additionally, fluorine atoms in the –CF₃ group form hydrogen bonds with MA⁺ cations, stabilizing MA sites and reducing V_{MA} density (Supplementary Fig. 13).” **Page 8, Line 210, Main text.**

7. “What does Supplementary Fig. 11-2 (Page 7 and Line 180) refer to?”

Response: We thank the reviewer for pointing out this mistake. The correct figure should be “**Supplementary Fig. 12b**” instead of “**Supplementary Fig. 11-2**”. We have corrected this error in

the revised manuscript to avoid confusion. **Page 7, Line 175, Main text.**

8. “The formula for calculating trap density on Page 10 and Line 271 has formatting issues. The authors should revise the formula and recalculate the trap density for perovskite SCNWAs.”

Response: We sincerely thank the Reviewer for pointing out the formatting issues in the formula for calculating trap density on Page 10, Line 271, as well as for highlighting the importance of correctly calculating this key parameter. According to the standard approach reported in the literature (*Nat. Electron.* 2018, 1, 404; *Nat. Commun.* 2025, 16, 753; *Nat. Commun.* 2018, 9, 3880), the correct formula for calculating trap density using the SCLC method is:

$$N_t = \frac{2\epsilon_r \epsilon_0 V_{TFL}}{qL^2}$$

Where ϵ_r is the relative dielectric constant (the value is 25.5, *Science*. 2015, 347, 519), ϵ_0 is the vacuum permittivity (the value is 8.85×10^{-14} F/cm), q is the elementary charge (the value is 1.60×10^{-19} C), V_{TFL} is the trap-filled limit voltage, and L is the electrode spacing (the value is 3.5×10^{-3} cm).

- After correcting all of these issues and carefully recalculating the trap density with the correct formula, we now obtain $N_t = 3.08 \times 10^{12} \text{ cm}^{-3}$ for the unpassivated MAPbBr₃ SCNWAs and $N_t = 1.13 \times 10^{12} \text{ cm}^{-3}$ for the MTFA-passivated MAPbBr₃ SCNWAs. These values show good agreement, in both trend and order of magnitude, with the trap densities independently extracted from TRPL measurements using the PEARS fitting tool under different excitation fluences, further confirming the effective passivation achieved by the MTFA treatment.

Revised Statement:

- The following content: “The trap density (N_t) was calculated using the formula $V_{TFL} = qN_t L^2 / 2\epsilon$, where q is the elementary charge, L is the thickness, and ϵ is the permittivity. The resulting trap density of $5.37 \times 10^8 \text{ cm}^{-3}$ surpasses that of most bulk MAPbBr₃ single crystals, highlighting the superior quality of the MTFA-treated SCNWAs” was replaced by “The N_t was calculated using the formula $V_{TFL} = qN_t L^2 / 2\epsilon_0 \epsilon_r$, where ϵ_r is the relative dielectric constant (25.5), ϵ_0 is the vacuum permittivity (8.85×10^{-14} F/cm), q is the elementary charge (1.60×10^{-19} C), V_{TFL} is the trap-filled limit voltage, and L is the electrode spacing (3.5×10^{-3} cm). The calculated trap

density is $1.13 \times 10^{12} \text{ cm}^{-3}$, which is comparable to that of many bulk MAPbBr₃ single crystals, indicating the optimization of SCNWAs quality by MTFE treatment.” Page 11, Line 301, Main text.

- The defect density values of Fig. 3d and e were revised.

9. “On Page 10, the same concept of trap density is denoted as N_t in the SCLC measurement (Line 270), but as $tDOS$ in TAS measurement (Line 281). Please ensure consistent labelling of this concept throughout the manuscript.”

Response: We thank the reviewer for pointing out the inconsistency in the labeling of trap density. We have now standardized the notation throughout the manuscript by consistently using N_t to represent trap density in both the SCLC and TAS measurements. This change helps to improve clarity and avoid confusion.

- The label of “trap stage density (NT)” was replaced by “trap density (N_t)”. Page 11, Line 278, Main text.
- The label of “trap density (N_t)” was replaced by “ N_t ”. Page 11, Line 301, Main text.
- The label of “trap density (N_t)” was replaced by “ N_t ”. Page 11, Line 304, Main text.
- The label of “trap density” was replaced by “ N_t ”. Page 12, Line 308, Main text.

10. “The authors employ hole-only and electron-only devices to calculate the trap density of perovskite SCNWAs. To enhance the reproducibility and transparency of the study, the corresponding experimental details should be provided in the Experimental section.”

Response: We thank the Reviewer for the careful reading and helpful suggestion. In response, we have added the corresponding experimental details to the Experimental section. We hope this revision addresses the reviewer’s concern.

Revised Statement:

- The corresponding experimental details: “All SCLC measurements were performed on hole-only devices with the structure Au/SCNWAs/Au, where Au electrodes (100 nm) were thermally evaporated onto both ends of the SCNWAs with an interelectrode spacing of approximately 35

μm . I–V curves were recorded using a Keysight B2901A source meter. To minimize the influence of ion migration, we adopted a scan rate of 10 V/s, which showed negligible hysteresis.” was added to the Experimental Section.

11. *“On Page 11 and Line 299, the XRD patterns should be referenced as Fig. 3h instead of Fig. 3g. In addition, the measured R, LDR, D^* and noise current at different biases are presented in Supplementary Fig. 18 instead of Supplementary Fig. 19. Please correct these references to ensure consistency with the figures provided.”*

Response: Thanks for your careful checks. We are very sorry for our carelessness. Based on your comments, we have corrected these figure references in the manuscript and supporting information.

To avoid any misinterpretation by the readers, we have made the following revisions:

- “MTFA-treated devices retained 98% of their initial photocurrent after over 4000 hours of air exposure, with no significant PbBr_2 peaks observed in XRD patterns (Fig. 3g).” was replaced by “As shown in Figure 3g, the MTFA-treated device retained 98% of the initial photocurrent after more than 4000 hours of air exposure, and no obvious PbBr_2 peak was observed in the XRD spectrum (Fig. 3h).” **Page 12, Line 337, Main text.**
- “This is further supported by contact angle measurements, which show a 35° increase in hydrophobicity for treated samples (Fig. 3h).” was replaced by “This is further supported by the contact angle measurements, with the inset of Fig. 3h showing a 35° increase in hydrophobicity for the treated sample.” **Page 13, Line 344, Main text.**
- The figure reference “Supplementary Fig. 18” was replaced by “Supplementary Fig. 21”. **Page 15, Line 392, Main text.**
- The figure reference “Fig S18” was replaced by “Supplementary Fig. 20”. **The ending part of “Preparation of MAPbX_3 -based SCNWAs for Photodetector Devices”, Page 4, Experimental Section.**

12. *“The active area has a significant effect on the calculated responsivity and detectivity of photodetectors. Based on the description in the Experimental section and Supplementary Fig 18,*

the active area appears to be underestimated, which potentially leading to overestimated device parameters. To ensure transparency and accuracy, the corresponding calculation process should be detailed in the Supporting Information.”

Response: We sincerely thank the reviewer for pointing out the importance of accurate area reporting in performance evaluation. We have carefully re-examined the device area description and provide the following clarifications and corrections:

(1) Clarification of Actual Area Used in Calculations:

We acknowledge that the description of the device area in the Supporting Information was inaccurate. Specifically, the previously stated value of $35 \times 5 \mu\text{m}^2$ was incorrect. In fact, the area used in our original calculations was based on $A = 35 \mu\text{m}$ (electrode spacing) \times (10 nanowires \times $5 \mu\text{m}$ width) = $1750 \mu\text{m}^2$.

However, upon further inspection and based on the scale bar in Supplementary Fig. 18, we realized that the actual width of a single nanowire is $10 \mu\text{m}$, not $5 \mu\text{m}$. Therefore, the correct device area should be $A = 35 \mu\text{m} \times (10 \text{ wires} \times 10 \mu\text{m width}) = 3500 \mu\text{m}^2$.

To provide more accurate evidence of the device geometry—especially the electrode spacing and nanowire distribution—we have added Fig. R7 in the revised Supplementary information. This figure includes the CAD design layout used for Au electrode fabrication and a low-magnification optical overview image of the device, which clearly shows the overall configuration and confirms the dimensions.

(2) Impact on Performance Metrics & Revision:

The previously reported Responsivity (R) and Specific Detectivity (D^*) were calculated based on the misreported device area. Given that:

$$R = \frac{I_{\text{light}} - I_{\text{dark}}}{L_{\text{light}} A}$$

and

$$D^* = \frac{R \sqrt{A f}}{I_{\text{noise}}}$$

We have revised the performance metrics:

- The **R** is recalculated to be **1660 A W⁻¹**, reflecting a factor of 2 correction.
- The **D*** is recalculated to be **3.9 × 10¹⁴ Jones**, reflecting a factor of $\sqrt{2}$ correction.

These changes have been updated accordingly in both the main text and Supplementary Information.

Importantly, we confirmed that this correction does not alter the main conclusions or the comparative trends presented in the manuscript. We sincerely thank the reviewer for the careful examination and for drawing our attention to this issue.

Fig. R7. (a) Photograph of the physical shadow mask, comprising a 10×50 array of electrodes for device fabrication. (b) Corresponding CAD layout of the full electrode array. (c) Detailed CAD design of a single electrode unit, showing channel width and contact dimensions. (d) Optical microscope image of a single device, showing the Au contacts and the bridging MAPbX₃ SCNWAs.

Revised Statement:

- The previously reported R value of “ 3320 A W^{-1} ” ***was replaced by*** “ 1660 A W^{-1} ”. Page 2, Line 48, Main text; Page 4, Line 115, Main text; Page 14, Line 375, Main text; Page 16, Line 432, Main text; Supplementary Table 1.
- The previously reported D^* value of “ $5.5 \times 10^{14} \text{ Jones}$ ” ***was replaced by*** “ $3.9 \times 10^{14} \text{ Jones}$ ”. Page 2, Line 47, Main text; Page 4, Line 115, Main text; Page 15, Line 391, Main text; Page 16, Line 431, Main text; Supplementary Table 2.
- The previously reported R value of “ 4847 A W^{-1} ” ***was replaced by*** “ 2424 A W^{-1} ”. Page 15, Line 394, Main text.
- “Supplementary Figure 20” ***was replaced by*** “Fig. R7”.
- “Lateral-type photoresistors were fabricated by depositing interdigital Au electrodes (100 nm thickness) onto the arrays via vacuum evaporation using a metal mask, with an inter-electrode gap of approximately 25 μm . The active area of the device is around $35 \times 5 \mu\text{m}^2$ for 10 nanowires covered by a pair of electrodes (Supplementary Fig. 20).” ***was replaced by*** “Lateral-type photoresistors were fabricated by depositing interdigital Au electrodes (100 nm thickness) onto the arrays via vacuum evaporation using a metal mask, with an inter-electrode gap of approximately 35 μm . The active area of the device is about $35 \times 10 \times 10 \mu\text{m}^2$, with 10 nanowires covered by a pair of electrodes (Supplementary Fig. 20).” **Experimental Section.**

13. *“There are many formatting issues in this manuscript that affect the readability of the article. I list some examples below, and the authors need to check the manuscript carefully and repeatedly.*

(i) On Page 6 and Line 157, “Landau-Levich regin” should be corrected “Landau-Levich regime”.

(ii) The key terms that appear for the first time need to be abbreviated. However, the authors provide abbreviations at the end of the manuscript, such as SCNWAs (on Page 13, Line 327) and DFT (on Page 14, Line 389).

(iii) It is necessary to provide the correct superscripts and subscripts of relevant parameters. For example, superscripts in D^ (Page 15, Line 393) and subscripts in Iupper and Ilower (Page 13, Lines 341-342).*

(iv) On Page 13, the responsivity for photodetector at 5 V and 10 V bias are $3320 A W^{-1}$ (Line 336) and $4847 A/W$ (Line 355), respectively. In order to improve the standardization of the manuscript, it is recommended that the units be revised to a uniform format.

(v) Recheck all references for accuracy and completeness.

(vi) The description is inconsistent with the label in Fig. 3f.”

Response: We sincerely thank the reviewer for the detailed comments regarding formatting and manuscript consistency. We have carefully revised the main text to correct all formatting issues and improve readability. Below, we address each specific point raised:

(i) The typo “Landau-Levich regin” was replaced by “Landau-Levich regime”. **Page 6, Line 156, Main text.**

(ii) All key terms that first appear in the main text and supporting information are now properly abbreviated upon first mention.

● The full term “single-crystal nanowire arrays” was replaced by the abbreviation “SCNWAs” because it was not its first appearance. **Page 14, Line 366, Main text.**

● The full term “density functional theory” was replaced by the abbreviation “DFT” because it was not its first appearance. **Page 16, Line 428, Main text.**

● The full term “MAPbX₃ crystal arrays” was replaced by the abbreviation “MAPbX₃ SCNWAs” because it was not its first appearance. **Page 16, Line 427, Main text.**

● The abbreviation “R2R” was replaced by the full term “roll-to-roll” because it only appears once. **Page 3, Line 88, Main text.**

- The full term “dynamic template-assisted (DTA)” was replaced by the abbreviation “DTA” because it was not its first appearance. **Page 4, Line 103, Main text; Page 5, Line 133, Main text.**
- The full term “methyl trifluoroacetate (MTFA)” was replaced by the abbreviation “MTFA” because it was not its first appearance. **Page 4, Line 109, Main text.**
- The full term “polydimethylsiloxane (PDMS)” was replaced by the abbreviation “PDMS” because it was not its first appearance. **Page 4, Line 104, Main text; Page 5, Line 133, Main text.**
- The full term “responsivity” was replaced by the abbreviation “R” because it was not its first appearance. **Page 15, Line 394, Main text.**
- The abbreviation “SEM” was replaced by the full term “Scanning Electron Microscope (SEM)” because it was first appearance. **Page 5, Line 124, Main text.**
- The abbreviation “EDS” was replaced by the full term “Energy Dispersive Spectrometer (EDS)” because it was the first appearance. **Page 5, Line 125, Main text.**
- The abbreviation “XRD” was replaced by the full term “X-ray Diffraction (XRD)” because it was first appearance. **Page 5, Line 128, Main text.**
- The full term “photoluminescence (PL)” was replaced by the abbreviation “PL” because it was not its first appearance. **Page 10, Line 272, Main text; Page 14, Supplementary Fig. 9, Supporting Information.**
- The abbreviation “XPS” was replaced by the full term “X-ray Photoelectron Spectroscopy (XPS)” because it was the first appearance. **Page 7, Line 192, Main text.**
- The abbreviation “FTIR” was replaced by the full term “Fourier-transform infrared spectroscopy (FTIR)” because it was first appearance. **Page 8, Line 193, Main text.**
- The full term “85% relative humidity” was replaced by the abbreviation “85% RH” because it was not its first appearance. **Page 4, Line 112, Main text; Page 13, Line 340, Main text; Page 13, Line 357, Main text; Page 15, Line 403, Main text; Page 16, Line 433, Main text.**
- The full term “photoluminescence quantum yield (PLQY)” was replaced by the abbreviation “PLQY” because it was not its first appearance. **Page 11, Line 281, Main text.**
- The “Space charge limited current” was replaced by the full term “Space charge limited current (SCLC)” because it was the first appearance. **Page 10, Line 259, Main text.**

- The abbreviation “D*” was replaced by the full term “Specific detectivity” because it was the first appearance. **Page 13, Line 354, Main text.**

(iii) All superscripts and subscripts in the text and supporting information have been carefully reviewed and corrected.

- The superscript in “D*” was replaced by “D^{*}”. **Page 14, Line 389, Main text.**
- The subscript in “I_{upper}” was replaced by “I_{upper}”. **Page 14, Line 381, Main text.**
- The subscript in “I_{lower}” was replaced by “I_{lower}”. **Page 14, Line 381, Main text.**
- The subscript in “N_T” was replaced by “N_T”. **Page 11, Line 278, Main text.**
- The subscript in “V_{TFL}” was replaced by “V_{TFL}”. **Page 11, Line 301, Main text.**

(iv) The units for responsivity have been standardized to the format of $A W^{-1}$ throughout the manuscript. Furthermore, we conducted a thorough review of the entire manuscript and supporting information to ensure consistency in all unit formatting and have made the necessary corrections where needed.

- The unit format of responsivity “A/W” was replaced by “ $A W^{-1}$ ”. **Page 15, Line 394, Main text.**
- The unit format of responsivity “ $\mu W/cm^2$ ” was replaced by “ $\mu W cm^{-2}$ ”. **Fig. 4a, Main text.**

(v) All references in both the main text and the Supporting Information have been thoroughly checked and revised to ensure completeness, accuracy, and consistency with the journal’s formatting guidelines.

(vi) We thank the reviewer for pointing out the inconsistency between the text and the label in Fig. 3f. In the original manuscript, we referred to the energy ranges 0.3–0.4 eV, 0.4–0.45 eV, and 0.45–0.5 eV as Band 1, 2, and 3, respectively, but the description did not clearly align with the Band I–III notations.

- “Thermal admittance spectroscopy (TAS) revealed a significant reduction in trap density across all defect levels in MTFAs-treated SCNWAs⁴¹ (Fig. 3f). (Band 1 (0.3–0.4 eV), associated with shallow traps, was reduced to ~10% of the control, consistent with suppressed bromide loss and aligning with DFT predictions. More strikingly, Bands 2 and 3, representing deeper-level defects, exhibited a nearly two-order-of-magnitude decrease in trap density (tDOS), underscoring MTFAs’s efficacy in passivating these more detrimental traps.” was replaced by “Thermal admittance spectroscopy (TAS) revealed a significant reduction in trap density across all defect levels in MTFAs-treated SCNWAs⁴¹. Band I (0.30–0.40 eV), associated with shallow traps,

showed a reduction to ~10% of the control level, consistent with suppressed bromide loss and aligning with DFT predictions. More notably, Band II (0.40–0.45 eV) and Band III (0.45–0.50 eV), corresponding to deeper-level defects, exhibited an approximately one-order-of-magnitude decrease in N_t , highlighting MTFFA's effectiveness in mitigating these more detrimental trap states." **Page 12, lines 308, Main text.**

Reviewer #2:

“In this manuscript, Mingjie Feng and colleagues present a novel blade-coating-based process for the large-area growth of perovskite microwire arrays, successfully fabricating high-quality single-crystalline perovskite microwire arrays. Furthermore, the authors employed MTF material to passivate defects within the single-crystalline microwire arrays, achieving an ultra-low defect state density of $5.37 \times 10^8 \text{ cm}^{-3}$. The resulting devices exhibit a remarkable responsivity of 3320 A/W and a specific detectivity of 5.5×10^{14} Jones, demonstrating superior optoelectronic performance compared to similar devices.

However, the manuscript requires further refinement in several aspects, with certain sections needing more detailed explanations. Most critically, we question the reported defect density and optoelectronic performance, which may not be as exceptional as claimed. We suspect potential issues in the calculations of defect density and optoelectronic performance presented in the paper. Therefore, we request that the authors address the following concerns thoroughly. The suitability of this manuscript for publication in Nature Communications will be evaluated based on the authors’ responses to these issues.”

Response: Thank you for understanding the importance of our work and for your constructive feedback. We carefully considered your comments and revised the manuscript accordingly. Please see our detailed response below.

1. *“On line 88, the authors introduce the abbreviation “R2R” for “roll-to-roll.” However, this term appears only once in the main text. We recommend removing the abbreviation to maintain clarity and consistency.”*

Response: We thank the Reviewer for the helpful suggestion. In accordance with the comment, we have removed the abbreviation “R2R” and now use the full term “roll-to-roll” to maintain clarity and consistency throughout the manuscript. Furthermore, we have carefully checked the entire manuscript and supporting information to ensure that all technical terms are properly written in full at their first appearance, with abbreviations introduced only when necessary:

- The abbreviation “R2R” **was replaced by** the full term “roll-to-roll” because it only appears once.

Page 3, Line 88, Main text.

- The full term “single-crystal nanowire arrays” was replaced by the abbreviation “SCNWAs” because it was not its first appearance. **Page 14, Line 366, Main text.**
- The full term “density functional theory” was replaced by the abbreviation “DFT” because it was not its first appearance. **Page 16, Line 428, Main text.**
- The full term “MAPbX₃ crystal arrays” was replaced by the abbreviation “MAPbX₃ SCNWAs” because it was not its first appearance. **Page 16, Line 427, Main text.**
- The full term “dynamic template-assisted (DTA)” was replaced by the abbreviation “DTA” because it was not its first appearance. **Page 4, Line 103, Main text; Page 5, Line 133, Main text.**
- The full term “methyl trifluoroacetate (MTFA)” was replaced by the abbreviation “MTFA” because it was not its first appearance. **Page 4, Line 109, Main text.**
- The full term “polydimethylsiloxane (PDMS)” was replaced by the abbreviation “PDMS” because it was not its first appearance. **Page 4, Line 104, Main text; Page 5, Line 133, Main text.**
- The full term “responsivity” was replaced by the abbreviation “R” because it was not its first appearance. **Page 15, Line 394, Main text.**
- The abbreviation “SEM” was replaced by the full term “Scanning Electron Microscope (SEM)” because it was first appearance. **Page 5, Line 124, Main text.**
- The abbreviation “EDS” was replaced by the full term “Energy Dispersive Spectrometer (EDS)” because it was the first appearance. **Page 5, Line 125, Main text.**
- The abbreviation “XRD” was replaced by the full term “X-ray Diffraction (XRD)” because it was first appearance. **Page 5, Line 128, Main text.**
- The full term “photoluminescence (PL)” was replaced by the abbreviation “PL” because it was not its first appearance. **Page 10, Line 272, Main text; Page 14, Supplementary Fig. 9, Supporting Information.**
- The abbreviation “XPS” was replaced by the full term “X-ray Photoelectron Spectroscopy (XPS)” because it was the first appearance. **Page 7, Line 192, Main text.**
- The abbreviation “FTIR” was replaced by the full term “Fourier-transform infrared spectroscopy (FTIR)” because it was first appearance. **Page 8, Line 193, Main text.**
- The full term “85% relative humidity” was replaced by the abbreviation “85% RH” because it

was not its first appearance. **Page 4, Line 112, Main text; Page 13, Line 340, Main text; Page 13, Line 357, Main text; Page 15, Line 403, Main text; Page 16, Line 433, Main text.**

- The full term “photoluminescence quantum yield (PLQY)” was replaced by the abbreviation “PLQY” because it was not its first appearance. **Page 11, Line 281, Main text.**
- The “Space charge limited current” was replaced by the full term “Space charge limited current (SCLC)” because it was the first appearance. **Page 10, Line 259, Main text.**
- The abbreviation “D*” was replaced by the full term “Specific detectivity” because it was the first appearance. **Page 13, Line 354, Main text.**

2. “On line 180, the authors reference Supplementary Fig. 11-2. However, this figure is not included in the supplementary information.”

Response: We appreciate the reviewer’s careful reading. The figure should be cited as “**Supplementary Fig. 11b**” instead of “**Supplementary Fig. 11-2**”. We have corrected this error in the revised manuscript to avoid confusion. **Page 7, Line 175, Main text.**

3. “On lines 341–342, the authors refer to “where Iupper and Ilower” without applying subscripts to the corresponding terms.”

Response: Thank you for pointing out the inconsistency on lines 341–342 regarding the missing subscripts for “Iupper” and “Ilower.” We have corrected this issue by properly formatting the terms with subscripts in the revised manuscript. In addition, we carefully reviewed the entire manuscript and supporting information for similar formatting issues involving subscripts and superscripts. All such instances have been corrected to ensure consistency and clarity throughout the text.

- The superscript in “D*” was replaced by “D^{*}”. **Page 14, Line 389, Main text.**
- The subscript in “Iupper” was replaced by “I_{upper}”. **Page 14, Line 381, Main text.**
- The subscript in “Ilower” was replaced by “I_{lower}”. **Page 14, Line 381, Main text.**
- The subscript in “NT” was replaced by “N_T”. **Page 11, Line 278, Main text.**
- The subscript in “VTFL” was replaced by “V_{TFL}”. **Page 11, Line 301, Main text.**

4. *“On line 353, the authors state, “Supplementary Fig. 18 shows the measured R, LDR, ...” However, the corresponding figure in the supplementary information is actually Supplementary Fig. 19. We recommend that the authors carefully review the manuscript to ensure the accuracy of all figure references.”*

Response: We appreciate your careful review and helpful observation regarding the incorrect figure reference on line 353. The citation has been corrected — “Supplementary Fig. 18” is now accurately referred to as “Supplementary Fig. 21” in the revised manuscript. Moreover, we have conducted a thorough check of all figure references in both the manuscript and the supporting information to ensure their accuracy and consistency throughout.

- The figure reference “Supplementary Fig. 18” was replaced by “Supplementary Fig. 21”. **Page 15, Line 392, Main text.**
- The figure reference “Fig S18” was replaced by “Supplementary Fig. 20”. **The ending part of 'Preparation of MAPbX₃-based SCNWAs for Photodetector Devices, Page 4, Experimental Section, Supporting Information.**
- To avoid any misinterpretation by the readers, we have made the following revisions: “MTFA-treated devices retained 98% of their initial photocurrent after over 4000 hours of air exposure, with no significant PbBr₂ peaks observed in XRD patterns (Fig. 3g).” was replaced by “As shown in Figure 3g, the MTFA-treated device retained 98% of the initial photocurrent after more than 4000 hours of air exposure, and no obvious PbBr₂ peak was observed in the XRD spectrum (Figure 3h).” **Page 12, Line 337, Main text.**
“This is further supported by contact angle measurements, which show a 35° increase in hydrophobicity for treated samples (Fig. 3h).” was replaced by “This is further supported by the contact angle measurements, with the inset of Figure 3h showing a 35° increase in hydrophobicity for the treated sample.” **Page 13, Line 344, Main text.**

5. *“In Figure 1e, why does the non-microwire region exhibit a certain level of lead element presence, while the bromine element is distinctly more concentrated in the microwire region? The authors should provide a detailed explanation for this observation.”*

Response: We thank the reviewer for their insightful comment regarding the apparent elemental

distribution differences between Pb and Br in Figure 1e. Specifically, the presence of lead signals in the non-microwire regions and the spatial confinement of bromine signals to the microwire regions can be rationalized by considering both the measurement conditions and elemental properties.

(1) EDS Voltage Selection and Its Impact on Elemental Detection:

The elemental mapping in Figure 1e was acquired using an accelerating voltage of 10 keV. This relatively low voltage was intentionally selected to avoid beam-induced degradation of the MAPbBr₃ single-crystalline nanowires during prolonged imaging. Based on our prior observations, higher voltages (≥ 15 keV) or long exposure times can lead to local heating and structural damage such as fragmentation or delamination of the nanowires. Therefore, a conservative imaging condition was chosen to preserve structural integrity, albeit at the expense of sensitivity to lighter elements like Br.

(2) Signal Discrepancy between Pb and Br: Role of Fluorescence Yield and Detection Efficiency:

The differing visibility of Pb and Br in the non-microwire regions can be largely attributed to the inherent differences in their EDS signal generation efficiency—most notably, the fluorescence yield (ω) of their characteristic X-ray lines. Although our EDS mapping was conducted at 10 keV, which is well below the optimal excitation voltage for Pb L α lines (~ 13.7 keV), strong Pb M α (~ 2.34 keV) signals are still observed across the sample (*J. Vac. Sci. Technol. A.* 2016, 34, 041501; *Mikrochim. Acta.* 2002, 138, 225). This is primarily due to the high fluorescence yield of Pb M α emissions ($\omega \approx 0.0327$), which enables detectable X-ray output even under suboptimal excitation conditions (*J. Phys. Chem. Ref. Data.* 1994, 23, 339). By comparison, the fluorescence yield of Br L α (~ 1.48 keV) is substantially lower ($\omega \approx 0.00032$), nearly two orders of magnitude less than that of Pb, resulting in inherently weaker signals under the same conditions. Pb's strong M α emission, on the other hand, allows for detectable signals even from residual material in non-microwire regions—likely originating from precursor overspray or diffusion during printing. However, the Pb intensity in these regions remains significantly lower than in the microwire channels, consistent with unintentional trace deposition rather than targeted patterning.

(3) Follow-Up EDS Measurements at Higher Voltages (15 and 20 keV):

To clarify this issue, we performed additional EDS mapping at higher accelerating voltages of 15

and 20 keV, enabling efficient excitation of Br K α lines and enhancing detection sensitivity. As shown in Figure R8, both the 15 keV (Figure R8a) and 20 keV (Figure R8c) maps reveal a more comprehensive Br distribution. Notably, faint but discernible Br signals begin to emerge in the non-microwire regions at 15 keV and become more evident at 20 keV. This progressive appearance strongly supports the hypothesis that the initial lack of Br signal at 10 keV was due to insufficient excitation energy, rather than a true absence of bromine. These results reinforce the conclusion that both Pb and Br may be present at low levels outside the microwires due to minor precursor residue, and their detectability is strongly voltage-dependent. We adjusted the contrast of Figure R8 so that the signal in the non-microwire region can be seen more clearly on computer screens with different resolutions.

Fig. R8 (a) EDS mapping at 15 keV showing Br (green) and Pb (cyan) elemental distributions. (b) Corresponding SEM image of the SCNWAs (scale bar: 10 μm). (c) EDS mapping at 20 keV of the same region, showing increased signal intensity and enhanced Br detectability in the non-microwire regions.

Revised Statement:

- “Fig. 1e” was replaced by “Fig. R8a”.

6. *“It is generally understood that the selective trapping of one type of carrier and the release of another at the surface of microwires/nanowires is a critical physical process enabling multiplied responsivity in one-dimensional devices. Consequently, surface defects are considered a key factor in achieving high multiplication responses. However, the authors report maintaining exceptionally high responsivity despite passivating surface defects. The authors should clarify why such high responsivity persists post-passivation. Additionally, the manuscript does not present data comparing the device’s photoresponsivity before and after MTEA passivation. We*

strongly recommend that the authors provide these data.”

Response: We sincerely thank the Reviewer for this insightful comment and fully agree with the understanding that surface carrier trapping and selective release are key physical mechanisms contributing to the gain (G) in one-dimensional photodetectors. As the Reviewer rightly pointed out, surface defects—especially those capable of selectively trapping one type of carrier (typically holes) while allowing the other (typically electrons) to drift freely—can give rise to pronounced carrier multiplication effects in one-dimensional photodetectors such as microwires and nanowires. In such systems, the trapped carriers induce a persistent internal electric field or band bending at the surface, which prolongs the lifetime of mobile carriers and leads to significant photoconductive gain. This gain mechanism is particularly prominent in low-dimensional structures where surface-to-volume ratios are high, making surface states a dominant factor in carrier dynamics and responsivity enhancement (*Nano Lett.* 2007, 7, 1003).

The photodetectors are operated in a photoconductive mode, in which the photoconductive gain is governed by the ratio of the carrier lifetime (τ_{lt}) to the carrier transit time (τ_{tt}). While surface trapping effects can contribute to longer effective carrier lifetimes and thereby enhance gain, it is important to note that the gain also depends on the carrier mobility (μ), the applied bias voltage (V), and the device geometry (specifically the sample thickness D), as expressed by (*Nat. Commun.* 2015, 6 8724; *Adv. Mater.* 2015, 27, 1912; *Adv. Mater.* 2021, 33, 2006691):

$$G = \frac{\tau_{lt}}{\tau_{tt}} = \frac{\tau_{lt}}{D^2/\mu V}$$

In our case, the applied voltage was V=5 V and the channel length was D=35 μm . Carrier lifetimes were extracted from the decay phase of transient photocurrent measurements, yielding τ_{lt} =0.0368 ms for the passivated sample and τ_{lt} =0.0761 ms for the unpassivated one (Fig. R9a). The carrier mobilities were extracted from the space-charge-limited current (SCLC) region of the dark J–V curves by using the Mott-Gurney law: $\mu=8JL^3/9\epsilon_0\epsilon_r V^2$ (*Nat. Commun.* 2023, 14, 839; *Nat. Commun.* 2021, 12, 6603.). where J is the current density, ϵ_r is the relative dielectric constant, ϵ_0 is the vacuum permittivity, and V is the voltage within the trap-free Child region of the current–J–V curve. The calculated mobilities are 191.9 $\text{cm}^2 \text{V}^{-1} \text{s}^{-1}$ for the unpassivated sample and 512.6 $\text{cm}^2 \text{V}^{-1} \text{s}^{-1}$ for the passivated sample. This significant enhancement in carrier mobility is attributed to the effective

surface defect passivation achieved by MTFE treatment, which chemically passivates undercoordinated Pb^{2+} ions (shallow traps) and Pb–Pb dimers (deep traps), thereby reducing scattering centers and improving charge transport. Using these parameters, we calculate the photoconductive gain to be approximately 5961 for the unpassivated device and 7679 for the MTFE-passivated device. These results suggest that, although defect passivation may slightly reduce the carrier lifetime by eliminating trap-assisted gain channels, the substantial enhancement in carrier mobility more than offsets this effect, leading to an overall increase in photoconductive gain. This trend is consistent with previous reports on surface-passivated perovskite single crystals, where improved carrier transport following surface modification led to higher photoresponsivity, despite the suppression of trap-assisted gain pathways (*Sci. Adv.* 2021, 7, eabc8844; *Adv. Mater.* 2023, 35, 2210016; *J. Phys. Chem. Lett.* 2019, 10, 786.).

In response to the reviewer’s suggestion, Fig. R9b presents the responsivity data measured for the unpassivated MAPbBr_3 SCNWAs device. As shown, the unpassivated device exhibits a high responsivity (1372 A W^{-1}) at low light intensity due to trap-induced gain but suffers from a significant decline in responsivity at higher intensities ($>10^{-3} \text{ W cm}^{-2}$), which marks the onset of gain saturation and increased recombination losses. In contrast, the MTFE-passivated device (data shown in Fig. 4b, main text) exhibits an even higher responsivity of 1660 A W^{-1} under low illumination. This observation suggests that, rather than trap-assisted gain, the improvement in carrier mobility plays a more dominant role in enhancing the photodetector performance after passivation.

Fig. R9. (a) Transient photocurrent decay curves of MAPbBr₃ SCNWAs with and without MTFA treatment under pulsed illumination. Extracted biexponential fitting parameters (τ_1 , τ_2 , and average decay time τ_{decay}) are shown in the inset table. (b) Responsivity as a function of incident light intensity for the unpassivated device (W/O MTFA).

7. *“On line 214, the authors claim that binding energy calculations demonstrate that TFA preferentially adheres to the crystal surface. Could the authors clarify the methodology and evidence supporting this conclusion? A detailed explanation of how the binding energy calculations substantiate this preference is needed.”*

Response: We thank the reviewer for pointing out the need for clarification regarding the basis of our conclusion about the preferential surface interaction of TFA⁻. Here, we provide a clearer explanation of the computational methodology and the supporting evidence used to substantiate this claim.

(1) Methodology for Binding Energy Calculations:

In our DFT calculations, we systematically evaluated the binding energies of three representative surface passivation ligands—Br⁻, CH₃COO⁻ (Ac⁻), and CF₃COO⁻ (TFA⁻)—on two dominant crystallographic terminations of MAPbBr₃ (PbBr and MABr surfaces). The computed binding energies (E_{bind}) were defined as (*Chem. Eng. J.* 2022, 440,135974):

$$E_{\text{bind}} = E_{\text{surface+ligand}} - E_{\text{surface}} - E_{\text{ligand}}$$

A more negative E_{bind} indicates stronger interaction and more favorable surface adhesion (*Energy Environ. Sci.* 2021, 14, 1429; *Adv. Mater.* 2024, 36, 2400347). Our results revealed that TFA⁻ exhibited the **most negative binding energy on both terminations**, surpassing those of Br⁻ and Ac⁻ (Fig. 2c, Main text), which suggests a stronger thermodynamic driving force for TFA⁻ to bind to the perovskite surface.

(2) Supporting XPS Evidence:

To further corroborate the theoretical findings, we conducted XPS measurements on MAPbBr₃ SCNWAs treated with Br⁻, Ac⁻, and TFA⁻, respectively (Fig. R10). The results revealed that:

- The Pb 4f binding energy of TFA-treated samples showed the largest shift toward **lower binding energy** compared to the other two, indicating stronger electron donation to Pb²⁺ and hence stronger Pb–O coordination.

- Simultaneously, the N 1s signal exhibited the most significant **shift toward higher binding energy** after TFA treatment, suggesting hydrogen bond formation or electrostatic interaction between MA^+ and the carboxylate oxygen of TFA^- .

XPS results provide experimental evidence that TFA exhibits dual-site interaction—coordinating with surface Pb^{2+} while simultaneously engaging MA^+ through hydrogen bonding. This dual interaction is more pronounced in TFA^- compared to the other ligands, aligning with the calculated binding energies that suggest a stronger surface affinity.

Fig. R 10. XPS spectra of (a) Pb 4f and (b) N 1s for MAPbBr_3 SCNWAs treated with Br^- , Ac^- , and TFA^- , respectively.

8. *“The experimental details require further elaboration. For instance, on line 228, the authors mention etching the microwire arrays. We recommend that the authors include a detailed description of the etching process in the experimental section to enhance the reproducibility and clarity of the study.”*

Response: Thank you for your valuable suggestion regarding the need for more detailed experimental descriptions. As you correctly pointed out, the etching process for the microwire arrays mentioned on line 228 lacked sufficient detail. We have now revised the experimental section to include a comprehensive description of this process. Moreover, in response to your broader concern, we have carefully reviewed the entire experimental section and have added further explanations or clarifications wherever necessary — particularly in procedures that may have appeared vague or insufficiently described.

- The following content: **“The XPS depth profiling was performed using a VG EX 05 mini-beam**

ion gun with 1000 eV Ar⁺ ions at 400 μm beam spot diameter. The etching rate was maintained at ~0.1 nm/s, as calibrated by a SiO₂/Si reference sample.” was added to the Experimental Section.

- The following content: “The absolute photoluminescence quantum yield (PLQY) was measured according to the method published by de Mello et al. Samples were placed in a 30 cm Spectralon-coated sphere coupled to an Ava Spec-2048L spectrometer by an optical fiber. The spectra were recorded with an excitation wavelength of 405 nm and corrected for the spectral sensitivity of the setup, determined with the help of a calibrated Xe lamp (Hamamatsu L7810-02).” was added to the Experimental Section.
- The following content: “For stability evaluation, five identical devices were simultaneously aged in an artificial weathering chamber (LHL-114, Espec Corp.) under controlled conditions (25 °C, 85% RH). The measured parameters were averaged across all samples at each predetermined time interval.” was added to the Experimental Section.
- The following content: “Absorption spectra of MAPbBr₃ SCNWAs were recorded at room temperature using a UV-vis-NIR spectrophotometer (PerkinElmer Lambda 950) equipped with an integrating sphere accessory, scanning over the 450-650 nm spectral range.” was added to the Experimental Section.

9. *“In Figure 3c, the authors present TRPL data for the microwire arrays but do not provide specific values for the carrier lifetime of the material before and after MTFA treatment. We recommend that the authors include these quantitative values to clearly demonstrate the impact of MTFA passivation.”*

Response: We thank the reviewer for this valuable suggestion. We agree that providing the specific values for the carrier lifetime would strengthen our discussion on the impact of MTFA passivation. In response to this comment, we have summarized the carrier lifetime values (τ_1 , τ_2 , and τ_{avg}) for both before and after MTFA treatment in **Table R2**.

Table R2. The fitted carrier lifetimes of MAPbBr₃ SCNWAs with and without (W/O) MTFA modification were obtained from the TRPL spectra.

	W/O MTFA (58.9 nJ/cm ²)	W/O MTFA (84.2 nJ/cm ²)	W/O MTFA (114.1 nJ/cm ²)	With MTFA (58.9 nJ/cm ²)	With MTFA (84.2 nJ/cm ²)	With MTFA (114.1 nJ/cm ²)
--	--	--	---	---	---	--

A ₁	0.50	0.43	0.45	0.43	0.44	0.50
τ ₁ (ns)	17.27	25.37	20.49	20.73	18.69	17.27
A ₂	0.42	0.45	0.44	0.48	0.47	0.42
τ ₂ (ns)	105.63	198.88	149.50	148.84	132.57	74.89
τ _{avg} (ns)	91.23	180.03	133.64	134.63	119.29	91.23
Coefficient R2	0.99	0.99	0.99	0.99	0.99	0.99

Note: The TRPL decay is fitted by a bi-exponential equation: $y=A_1 \exp\left(-\frac{x}{\tau_1}\right)+A_2 \exp\left(-\frac{x}{\tau_2}\right)$, where parameters A₁ and A₂ are the amplitude fraction for each decay component, τ₁ and τ₂ represent the time constant of the two types of decay. The average lifetime (τ_{avg}) can be calculated with the equation: $\tau_{avg} = \frac{(A_1\tau_1^2+A_2\tau_2^2)}{A_1\tau_1+A_2\tau_2}$.

We observe that MTFA-treated MAPbBr₃ microwire arrays exhibit significantly longer average carrier lifetimes under low excitation fluence (τ_{avg} = 134.63 ns with MTFA vs. 91.23 ns without MTFA at 58.9 nJ/cm²), which indicates effective suppression of nonradiative surface recombination by passivation. As the excitation fluence increases, the average lifetime of both unpassivated and passivated samples shows different trends. For the unpassivated sample, the lifetime first increases (from 91.23 ns to 180.03 ns) due to progressive trap filling, which reduces the influence of defect-assisted recombination. However, at the highest fluence (114.1 nJ/cm²), the lifetime decreases again (133.64 ns), suggesting the onset of bimolecular recombination or higher-order Auger recombination. In contrast, for the MTFA-passivated sample, the lifetime decreases steadily with increasing excitation fluence (from 134.63 ns to 119.29 ns and 91.23 ns), without the initial rise observed in the unpassivated case. This is because after effective defect passivation, the trap saturation effect is negligible, and the higher carrier density under strong excitation is also more likely to activate bimolecular recombination or possible Auger recombination, resulting in faster decay kinetics. This behavior has been reported in the preparation and performance studies of high-quality perovskite-based thin film materials (*Phys. Chem. Chem. Phys.* 2020, 22, 28345; *Adv. Energy Mater.* 2021, 11, 2003489; *Adv. Energy Mater.* 2020, 10, 1904134).

Revised Statement:

- We have added Table R2 to the Supplementary Information, where it can be seen in Supplementary Table 4.

- The following content: “In addition, we calculated the carrier lifetimes of the samples before and after MTFa treatment (Supplementary Table 4). The MAPbBr₃ microwire arrays treated with MTFa exhibited significantly extended average carrier lifetimes at low excitation energy densities ($\tau_{\text{avg}}=134.6$ ns, compared to 91.2 ns for the untreated sample at 58.9 nJ/cm², Supplementary Table 3), indicating that non-radiative recombination was suppressed. Interestingly, we found that τ_{avg} of the unpassivated sample first increased and then decreased with increasing energy density, which was due to trap filling followed by bimolecular or Auger recombination, while that of the passivated sample showed a steady downward trend, which was consistent with the suppression of trap saturation. These behaviors are consistent with previous reports on high-quality perovskite films.” was added to the main text on Page 11, Line 281.

10. *“In Supplementary Fig. 15, the authors present J-V curves (reverse and backward sweeps) for electron-only devices under different scan rates. However, several issues require clarification. First, the authors do not specify the detailed device structure of the electron-only devices. Second, the y-axis is labeled in current units, so the curve should be named I-V curve. Furthermore, the presented data clearly indicate a pronounced hysteresis effect in the devices, which is only partially mitigated at higher scan rates. We recommend that the authors address these points by providing the device structure, correcting the name, and discussing the observed hysteresis in greater detail.”*

Response: We sincerely thank the reviewer for their careful examination of Supplementary Fig. 15 and their constructive suggestions. Below, we address each point raised:

(1) Device structure clarification:

We agree that the device structure was not explicitly stated in the original supplementary figure caption. To clarify, the device described in Supplementary Fig. 15 was mistakenly labeled as an electron-only structure. Contrary to the initial description, the device used in this measurement is actually a hole-only lateral structure (Au/SCNWAs/Au), consistent with the device architecture presented in Fig. 3d–e of the main text. We sincerely apologize for this oversight in labeling.

We have updated the caption of Supplementary Fig. 15 to correct this information and have included the fabrication details of the hole-only device in the Experimental Section.

(2) Y-axis labeling and curve naming:

We acknowledge the reviewer's comment regarding the y-axis being labeled in current (A) rather than current density (mA cm^{-2}). We have revised the curve name to “**I–V curves**” accordingly to maintain consistency with the axis label.

(3) Hysteresis discussion:

Owing to the mixed ionic–electronic conduction behavior inherent to perovskite materials, the accuracy of charge trap density extracted from SCLC measurements is significantly influenced by external measurement parameters, including scan rate, scan direction, and temperature (*Nat. Commun.* 2020, 11, 4023; *ACS Energy Lett.* 2020, 5, 376). This phenomenon primarily originates from the migration of mobile ions (such as MA^+ or Br^-) under electric fields (*Adv. Energy Mater.* 2015, 5, 1500615; *Energy Environ. Sci.* 2015, 8, 2118). To minimize these non-ideal contributions, we measured the I–V characteristics of hole-only devices under various scan rates, ranging from 10 mV s^{-1} to 10 V s^{-1} (see Supplementary Fig. 15). At low scan rates, pronounced hysteresis is observed due to sufficient time for ion redistribution and trap-state relaxation. However, as the scan rate increases, these slow-response processes are effectively suppressed, leading to a marked reduction in hysteresis (*J. Phys. Chem. C.* 2019, 7, 4029; *ACS Energy Lett.* 2022, 7, 3235). When the scan rate reaches $10,000 \text{ mV/s}$, the hysteresis effect becomes nearly negligible, as mobile ions no longer have sufficient time to respond to the external electric field. Consequently, the influence of ion migration on the SCLC measurement is effectively minimized (*Adv. Energy Mater.* 2021, 11, 2101447).

Revised Statement:

- Figure caption: “Supplementary Fig. 15 J–V curves of electron-only devices measured with different scan rates at room temperature (reverse and backward sweep).” was replaced by “Supplementary Fig. 16 I–V curves of hole-only lateral devices (Au/SCNWAs/Au) measured at room temperature under different scan rates (reverse and forward sweeps).”
- The following content: “We attribute this behavior to the mixed ionic–electronic conduction nature of perovskites, where ion migration under an electric field can distort SCLC measurements. At low scan rates, mobile ions have sufficient time to redistribute, resulting in significant hysteresis. By contrast, increasing the scan rate effectively suppresses ion movement

and trap-state relaxation. When the scan rate reaches 10,000 mV/s, the hysteresis effect becomes nearly negligible, as mobile ions no longer have sufficient time to respond to the external electric field.” was added to Page 11, Line 294, Main text.

11. “We have significant concerns regarding the authors' calculation of defect state density based on the SCLC theory. First, the formula presented on line 271 contains clear errors: (i) relevant terms lack appropriate subscripts, and (ii) the actual formula for calculating defect state density appears to be incorrect. The correct formula should be (Single-crystalline layered metal-halide perovskite nanowires for ultrasensitive photodetectors. Nat Electron 1, 404–410 (2018)). Furthermore, assuming a dielectric constant of approximately 20 for MAPbBr₃, and using the authors' provided electrode spacing of 25 μm and VTFL of 1.34 V, we estimate the defect density to be approximately $5 \times 10^{12} \text{ cm}^{-3}$. This value is significantly higher than the reported $8.72 \times 10^9 \text{ cm}^{-3}$. We request that the authors provide a detailed explanation for this discrepancy, including a thorough review of their calculation methodology and assumptions.”

Response: We sincerely thank the reviewer for the careful reading and constructive comments regarding the SCLC-based calculation of defect state density. We have carefully reviewed our formula, assumptions, and calculations as suggested. Our response is provided in two parts below:

- Regarding the formula and notation in Line 271, we agree that the notations were not clearly presented. In the revised manuscript, we have corrected the formula for trap density N_t and added appropriate subscripts to clarify the terms involved. According to the standard approach reported in the literature (*Nat. Electron.* 2018, 1, 404; *Nat. Commun.* 2025, 16, 753; *Nat. Commun.* 2018, 9, 3880), the correct formula for calculating trap density using the SCLC method is:

$$N_t = \frac{2\epsilon_r\epsilon_0 V_{TFL}}{qL^2}$$

Where ϵ_r is the relative dielectric constant (the value is 25.5, *Science* 2015, 347, 519), ϵ_0 is the vacuum permittivity (the value is $8.85 \times 10^{-14} \text{ F/cm}$), q is the elementary charge (the value is $1.60 \times 10^{-19} \text{ C}$), V_{TFL} is the trap-filled limit voltage, and L is the electrode spacing (the value is $3.5 \times 10^{-3} \text{ cm}$). This corrected formula, with the appropriate subscripts now properly included, has been clearly presented in the main text, and the corresponding section has been updated

accordingly.

- Regarding the discrepancy in the calculated value of trap density, we appreciate the reviewer's re-calculation and careful comparison. After re-examining our original calculation, we identified that the previously reported low trap density ($8.72 \times 10^9 \text{ cm}^{-3}$ and $5.37 \times 10^8 \text{ cm}^{-3}$) resulted from a miscalculation arising from an incorrect handling of the unit conversion for the electrode spacing during the computation (while L was correctly written as $3.5 \times 10^{-3} \text{ cm}$, in the calculation step L^2 was inadvertently treated as $12.25 \times 10^{-3} \text{ cm}^2$ instead of the correct $12.25 \times 10^{-6} \text{ cm}^2$), which led to an underestimation of the trap density by approximately three orders of magnitude. Based on the correct formula, we now obtain $N_t = 3.08 \times 10^{12} \text{ cm}^{-3}$ for the unpassivated MAPbBr₃ SCNWAs and $N_t = 1.13 \times 10^{12} \text{ cm}^{-3}$ for the MTFE-passivated MAPbBr₃ SCNWAs. These values show good agreement, in both trend and order of magnitude, with the trap densities independently extracted from TRPL measurements using the PEARS fitting tool under different excitation fluences, further confirming the effective passivation achieved by the MTFE treatment. We have corrected the value in the manuscript accordingly and revised all associated discussion to reflect the corrected trap density. We sincerely apologize for this oversight and thank the reviewer for pointing it out. We hope that the revised calculation and clarification have now addressed the reviewer's concern.

Revised Statement:

- The following content: “The N_t was calculated using the formula $V_{TFL} = qN_tL^2/2\epsilon$, where q is the elementary charge, L is the thickness, and ϵ is the permittivity. The resulting trap density of $5.37 \times 10^8 \text{ cm}^{-3}$ surpasses that of most bulk MAPbBr₃ single crystals, highlighting the superior quality of the MTFE-treated SCNWAs” was replaced by “The trap density (N_t) was calculated using the formula $V_{TFL} = qN_tL^2/2\epsilon_0\epsilon_r$, where ϵ_r is the relative dielectric constant (25.5), ϵ_0 is the vacuum permittivity ($8.85 \times 10^{-14} \text{ F/cm}$), q is the elementary charge ($1.60 \times 10^{-19} \text{ C}$), V_{TFL} is the trap-filled limit voltage, and L is the electrode spacing ($3.5 \times 10^{-3} \text{ cm}$). The calculated trap density is $1.13 \times 10^{12} \text{ cm}^{-3}$, which is comparable to that of many bulk MAPbBr₃ single crystals, indicating the optimization of SCNWAs quality by MTFE treatment.” **Page 11, Line 301, Main text.**
- The defect density values of Fig. 3d and e were revised.

12. *“The effective area of the device is critical to the accuracy of the reported device performance metrics. In the Experimental Section, the authors state that the distance between device electrodes is approximately 25 μm . Based on the scale bar provided in Supplementary Fig. 18, we estimate the width of a single microwire to be approximately 10 μm . Consequently, the effective area of a single microwire in a device is approximately 250 μm^2 . Given that each device contains 10 microwires, the total effective area of a device is approximately 2500 μm^2 . However, the authors report a device area of $35 \times 5 \mu\text{m}^2$, which is 15 times smaller than the estimated effective area. Could the authors clarify how the $35 \times 5 \mu\text{m}^2$ area was determined? If the device area has been miscalculated, the reported responsivity (R) and specific detectivity (D^*) must be recalculated. We request a detailed explanation of the area calculation and, if necessary, a revision of the performance metrics based on the correct effective area.”*

Response: We thank the reviewer for carefully analyzing the effective device area and for pointing out the inconsistency.

(1) Clarification of Area Determination:

We take this comment very seriously and have carefully re-examined the relevant description. We found that the originally stated value of “ $35 \times 5 \mu\text{m}^2$ ” in the Supplementary Information was indeed a writing error. In fact, all our previous calculations were carried out using the formula: $A = 35 \mu\text{m}$ (length) \times (5 μm width \times 10 wires), where the electrode spacing is 35 μm and the metal contact spans 10 parallel microwires. Upon further careful checking, however, we realized that the width of each microwire was underestimated. The actual width is $\sim 10 \mu\text{m}$ rather than 5 μm , which means that the correct device area should be:

$$A = 35 \mu\text{m} (\text{length}) \times (10 \mu\text{m} \text{ width} \times 10 \text{ wires}) = 3500 \mu\text{m}^2$$

To more accurately demonstrate the structure of our device and clearly show the electrode spacing, we have included in Fig. R11 the original CAD layout used for fabricating the Au electrodes, along with a low-magnification optical image that displays the full device configuration.

(2) Impact on Device Metrics & Revision

The previously reported Responsivity (R) and Specific Detectivity (D^*) were calculated based on the misreported device area. Given that:

$$R = \frac{I_{\text{light}} - I_{\text{dark}}}{L_{\text{light}} A}$$

and

$$D^* = \frac{R \sqrt{Af}}{I_{\text{noise}}}$$

Accordingly, we have revised the performance metrics:

- The **R** is recalculated to be **1660 A/W**, reflecting a factor of 2 correction.
- The **D*** is recalculated to be **3.9×10^{14} Jones**, reflecting a factor of $\sqrt{2}$ correction.

These values have been updated in the revised manuscript and Table of Supplementary Information.

We carefully verified that this correction has no impact on the main conclusions or the comparative trends of the study. We appreciate the reviewer's careful attention to this important detail, which has helped improve the accuracy and clarity of our reported results.

Fig. R11. (a) Photograph of the physical shadow mask, comprising a 10×50 array of electrodes for device fabrication. (b) Corresponding CAD layout of the full electrode array. (c) Detailed CAD design of a single electrode unit, showing channel width and contact dimensions. (d) Optical microscope image of a single device, showing the Au contacts and the bridging MAPbX₃ SCNWAs.

Revised Statement:

- The previously reported R value of “ 3320 A W^{-1} ” was replaced by “ 1660 A W^{-1} ”. Page 2, Line 48, Main text; Page 4, Line 115, Main text; Page 14, Line 375, Main text; Page 16, Line 432, Main text; Supplementary Table 3.
- The previously reported D^* value of “ 5.5×10^{14} Jones” was replaced by “ 3.9×10^{14} Jones”. Page 2, Line 47, Main text; Page 4, Line 115, Main text; Page 15, Line 391, Main text; Page 16, Line 431, Main text; Supplementary Table 4.
- The previously reported R value of “ 4847 A W^{-1} ” was replaced by “ 2424 A W^{-1} ”. Page 15, Line 394, Main text.
- “Supplementary Figure 20” was replaced by “Fig. R11”.
- “Lateral-type photoresistors were fabricated by depositing interdigital Au electrodes (100 nm

thickness) onto the arrays via vacuum evaporation using a metal mask, with an inter-electrode gap of approximately 25 μm . The active area of the device is around $35 \times 5 \mu\text{m}^2$ for 10 nanowires covered by a pair of electrodes (Supplementary Fig. 20).” was replaced by “Lateral-type photoresistors were fabricated by depositing interdigital Au electrodes (100 nm thickness) onto the arrays via vacuum evaporation using a metal mask, with an inter-electrode gap of approximately 35 μm . The active area of the device is about $35 \times 10 \times 10 \mu\text{m}^2$, with 10 nanowires covered by a pair of electrodes (Supplementary Fig. 20).” **Experimental Section.**

We sincerely thank all reviewers for their valuable and constructive comments, which have been extremely important for improving the quality of this work. We also acknowledge that some elementary errors in the original manuscript were due to our own oversight, and we have carefully addressed and corrected them in the revised version.

Reviewers' comments:

Reviewer #1:

“The authors have addressed the questions raised by the reviewers. I have no further questions regarding this manuscript.”

Response: Thank you very much for your careful review and suggestions to improve our manuscript. We appreciate your great contributions..

Reviewer #2:

1. *“The author has provided a detailed response to our earlier queries, addressing several initial concerns about the article. To further enhance the reliability of the reported specific detectivity (D^*), we recommend including the instrumental baseline noise data in a relevant section of the paper. This addition would help confirm that the noise data presented in Figure 4d are attributable to the photodetector itself, rather than the instrumental noise background.”*

Response: We sincerely thank the reviewer for this insightful suggestion. To address this concern, we have measured and included the instrumental baseline noise under identical experimental conditions without connecting the device. The results are now provided below (**Fig. R2-1**). As shown, the instrumental noise level is at least one order of magnitude lower than the noise current recorded for our MAPbBr₃ single-crystal nanowire array photodetectors. This confirms that the noise data presented in Fig. 4d and Supplementary Fig. 21 originate from the photodetector itself, rather than from the instrumental background.

Fig. R2-1. The instrument noise floor.

Revised Statement:

- We have incorporated Fig. R2-1 into Supplementary Fig. 22.